# Oxygen and glucose supplementation increases rates of linear growth and cell proliferation in growth-restricted fetal sheep

Mariangel Varela[1], Daniel B. Chrisenberry[1], Eliza H. Johnson[1], Rosa I. Luna-Ramirez[1], Ayna R. Tracy[1], Sanya Kathuria[1] ⓘ, Weicheng Zhao[1], Miranda J. Anderson[1], Laura D. Brown[2] ⓘ and Sean W. Limesand[1] ⓘ

[1]*School of Animal and Comparative Biomedical Sciences, University of Arizona, Tucson, Arizona, USA*
[2]*Department of Pediatrics, University of Colorado Anschutz Medical Campus, Aurora, Colorado, USA*

Handling Editors: Richard Carson & Kathy Ruddy

The peer review history is available in the Supporting Information section of this article (https://doi.org/10.1113/JP290141#support-information-section).

**Mariangel Varela** received her master's degree in animal and comparative biomedical sciences at the University of Arizona. Under the supervision of Dr. Sean Limesand, her master's research project focused on developing an intervention strategy to improve growth rates in fetal lambs that were growth restricted due to placental insufficiency. Mariangel is currently working as a scientist in the immunoassay department of a medical diagnostics company where she analyses cancer proteins in human blood. She is most passionate about advancing collective knowledge throughout the scientific community and using her own skills for better solutions in the medical field.

**Abstract figure legend** Effects of sustained oxygen and glucose supplementation on growth and cellular proliferation in growth-restricted fetuses. This study evaluated whether supplemental oxygen and glucose could improve growth rates in sheep fetuses with established fetal growth restriction (FGR). Following the induction of placental insufficiency and FGR via mid-gestational environmental hyperthermia, fetuses were supplemented for 10 days with either oxygen and glucose (FOG) or air and saline (FAS) and compared to a control (CON) group. Linear growth, measured by fetal thoracic circumference growth rates, was similar between FOG and CON fetuses, both of which significantly exceeded FAS rates. In FOG fetuses pancreatic β-cell proliferation was restored to normal levels, and skeletal muscle satellite cell proliferation was significantly increased compared to FAS fetuses. These findings demonstrate that prenatal intervention with oxygen and glucose increases linear growth and promotes cellular proliferation in tissues critical for glucose regulation.

**Abstract** Placental insufficiency lowers oxygen and glucose concentrations in the fetus, which causes fetal growth restriction (FGR). Previously we showed that FGR fetuses chronically supplemented with oxygen and glucose (OG) have improved glucose tolerance. However growth was not evaluated. Here we test the hypothesis that sustained OG supplementation to an FGR fetus will increase linear growth rates, raise anabolic hormone concentrations and promote proliferation rates in pancreatic β-cells and skeletal muscle satellite cells, which are two tissues involved in glucose regulation. FGR was induced in sheep with environmental hyperthermia. FGR fetuses were chronically supplemented with either oxygen and glucose (FOG) or air and saline (FAS) for 10 days and compared to thermoneutral controls. Before supplementation both FOG and FAS fetuses had lower arterial oxygen, glucose, insulin and insulin-like growth factor 1 (IGF-1) concentrations compared to controls. The OG supplementation successfully increased $PaO_2$, glucose and IGF-1 concentrations, but amino acid concentrations were unaffected. On day 8 of supplementation glucose-stimulated insulin concentrations were higher in FOG fetuses than FAS fetuses, whose insulin secretion was dependent on $PaO_2$. Fetal thoracic circumference growth rates, which measure linear growth, for FOG and control fetuses were similar and greater than FAS rates. Although linear growth rates were normalized, body weights for FOG and FAS groups remained lighter than controls. However β-cell and satellite cell proliferation rates were greater in FOG fetuses compared with FAS fetuses. Treatment of FGR with OG supplementation, two substrates transported across the placenta by diffusion, represents an innovative approach to reverse physiological challenges prenatally.

(Received 15 September 2025; accepted after revision 5 February 2026; first published online 16 March 2026)

**Corresponding author** S. W. Limesand: Animal and Comparative Biomedical Sciences, The University of Arizona, 1650 E Limberlost Dr, Tucson, AZ 85719, USA. Email: limesand@arizona.edu

**Key points**

- Simultaneous supplementation of oxygen and glucose to fetuses with placental insufficiency successfully increased fetal arterial oxygen, glucose and IGF-1 concentrations.
- Prenatal oxygen and glucose supplementation restored linear growth rates in growth-restricted fetuses to levels similar to healthy controls, although overall body weight remained lower.
- Treatment increased the proliferation rates of pancreatic β-cells and skeletal muscle satellite cells, addressing two key tissues that typically show reduced growth in untreated FGR fetuses.
- By the eighth day of supplementation FGR fetuses demonstrated improved glucose-stimulated insulin secretion, indicating a functional recovery of pancreatic response.
- These findings suggest that targeting substrates transported across the placenta by diffusion – like oxygen and glucose – is an innovative approach to prenatally reverse the physiological challenges of placental insufficiency.

## Introduction

Fetal growth restriction (FGR) often results from placental insufficiency, leading to fetal nutrient and oxygen deficiencies (Pardi et al., 2002; van Uden & Tchirikov, 2023). Consequently fetal hypoxemia and hypoglycaemia increase norepinephrine and lower insulin and insulin-like growth factor 1 (IGF-1) concentrations, which slow fetal growth rates (Chang et al., 2019; Limesand & Rozance, 2017). Notably placental insufficiency and FGR also reduce pancreatic $\beta$-cell and skeletal muscle satellite cell proliferation rates (Limesand et al., 2005; Rozance et al., 2018; Soto et al., 2017; Yates et al., 2014). These factors contribute to perinatal FGR pathophysiology and are associated with the future risk of developing glucose intolerance in adulthood (Hales et al., 1991; Phillips, 1995; Van Assche et al., 1977; Whincup et al., 2008). Despite urgent need no clinically proven intra-uterine treatment exists for pregnancies with placental insufficiency-induced FGR (Baschat, 2010; McCowan et al., 2018; Spiroski et al., 2018).

Fetal oxygen and nutrient deprivation due to placental insufficiency indicate intrauterine supplementation of oxygen and nutrients as a viable intervention after diagnosis (Nicolaides et al., 1987; van Uden & Tchirikov, 2023). A variety of nutrient interventions aimed to treat FGR in humans and experimental animal models have failed to improve growth trajectories (Bruin et al., 2021; Macko et al., 2016; Rozance et al., 2009; Wai et al., 2018; Zalinska et al., 2023). However these supplementation approaches are mostly dependent on single nutrients that in some instances lead to more severe hypoxia and acidosis (Brown et al., 2011; Rozance et al., 2009; Tchirikov et al., 2018). We therefore conducted an experiment where we supplemented glucose and oxygen simultaneously to fetal sheep with FGR for 5 days (Camacho et al., 2022). Significant improvements were shown in fetal glucose-stimulated insulin secretion (GSIS) and central insulin sensitivity (Camacho et al., 2022). Nevertheless questions remain as to whether the combined oxygen and glucose treatment will increase growth rates of the fetus after the onset of FGR or whether other factors become limited with a longer duration.

To address the question of growth and to expand on our FGR intervention, we measured linear growth rates in FGR fetal sheep during 10 days of supplementation with oxygen and glucose. We sought to confirm improvements in GSIS, evaluate insulin secretion dependence on fetal oxygenation and measure glucose-potentiated arginine-stimulated insulin secretion (GPAIS) after a week of supplementation. In addition we measured fetal plasma concentrations of insulin, IGF-1 and amino acids to determine whether growth potential is limited by anabolic hormones or other nutrient substrates for growth. We also determined proliferation rates *in situ* for the pancreatic $\beta$-cells and skeletal muscle satellite cells, which are key cell populations within tissues that regulate glucose homeostasis.

## Materials and methods

### Ethical approval

Animal procedures were approved by the Institutional Animal Care and Use Committee and were conducted in compliance with the Guide for the Care and Use of Laboratory Animals at the University of Arizona Agricultural Research Center (PHS Animal Welfare Assurance # D16-00159 (A-3248-01) and USDA Animal Research Facility Registration # 86-R-0003). The university is accredited by the Association for Assessment and Accreditation of Laboratory Animal Care International (accreditation # 000163). Experimental details are reported in compliance with the Animal Research: Reporting of In Vivo Experiments (ARRIVE) guidelines 2.0 (Grundy, 2015; O'Halloran, 2024; Percie du Sert et al., 2020).

### Fetal sheep preparation

Pregnant Columbia-Rambouillet crossbred ewes (57 $\pm$ 8 kg) were transported to the laboratory at 35 $\pm$ 3 days of gestation age (dGA). Pregnancies were identified using ultrasonography, and ewes carrying singletons were enrolled, although two ewes with twin pregnancies were identified later at surgery and assigned randomly to each FGR group (Fig. 1). Pregnant ewes were 2–5 years of age with unknown parity. The ewes were assigned to groups using simple randomization methods (RAND() function in Excel). Pregnant ewes ($n = 26$) were exposed to elevated ambient temperatures (40°C for 12 h; 35°C for 12 h; dew point 22°C) from day 40 $\pm$ 1 to day 87 $\pm$ 2 of gestation (term 149 days) to create fetuses with placental insufficiency and FGR. Control fetuses (CON; $n = 11$) were from pregnant ewes that were maintained at 22°C and pair-fed to the average *ad libitum* feed intake of ewes in the FGR group throughout pregnancy. Ewes were fed Standard Bread Alfalfa Pellets (Sacate Pellet Mills, Inc. Maricopa AZ, USA) supplemented with trace minerals for sheep (Code #1131-Z, Maid Rite Feeds, Willcox, AZ, USA) and provided water and white salt *ad libitum*. Sample size was determined using power calculations for group mean differences of 30% for GSIS measurements ($\alpha = 0.05$, $\beta = 0.85$ [Limesand et al., 2007]).

All pregnant ewes at 119 $\pm$ 2 dGA underwent a surgical procedure to place indwelling vascular catheters (Tygon ND-100-80 Flexible Plastic Tubing) as previously described (Davis et al., 2020; Macko et al., 2016). Briefly the induction of anaesthesia occurred with an injection

of ketamine (16 mg/kg) and diazepam (0.2 mg/kg) into a jugular vein of the ewe. At induction the ewe received an intramuscular injection of penicillin (VetOne, Boise, ID, USA; 30k units/kg) as a broad-spectrum antibiotic and an intravenous injection of Ketofen (Zoetis, Kalamazoo, MI, USA; 2.2 mg/kg) as a non-steroidal anti-inflammatory drug. The ewe was intubated, and unconsciousness was maintained by inhalation of 1.5–3.5% isoflurane in oxygen. Fetal vascular catheters were placed. A growth device catheter (Tygon ND-100-80 Flexible Plastic Tubing; AAD04127) was attached to the spinal processes, and the nylon monofilament was tunnelled subcutaneously and attached immediately anterior to the xiphisternum (Mellor & Murray, 1982; Rumball et al., 2009). The growth device measured the linear growth rate of half of the fetal thoracic circumference (girth). Maternal vascular catheters and tracheal catheter were also placed in each ewe in the FGR group (Yates et al., 2012). Ewes were given postoperative analgesics intravenously (10 mg/kg/day phenylbutazone, VetOne Boise, ID, USA) for two additional days. At the end of the intervention (132 ± 2 dGA) the ewe and fetus were humanely killed with an overdose of pentobarbital sodium given intravenously through the maternal and fetal femoral vein (86 mg/kg, Euthasol; VirbacAnimal Health, Fort Worth, TX, USA). Death was confirmed by loss of heartbeat and

breath and with a bilateral pneumothorax as a secondary method.

## Experimental design

At 122 ± 2 dGA experimental interventions were started in FGR fetuses that were assigned at random to one of two experimental conditions: FGR fetuses treated with oxygen and glucose (FOG), or fetuses treated with air and saline (FAS). Six FGR fetuses were lost before surgery, two male fetuses were lost during the intervention and two male fetuses were excluded because their values did not meet the inclusion criteria for FGR, which require $PaO_2$ <19 mmHg and plasma glucose concentrations <0.85 mmol/l. This results in a total of 16 FGR fetuses for subsequent studies. For the FOG group ($n = 8$ fetuses: 2 males, 6 females) a maternal tracheal insufflation of 100% humidified oxygen and an intravenous dextrose (30% solution) infusion into the fetus were started. Fetal arterial $PaO_2$ and glucose were monitored at least twice daily and adjusted by titrating maternal oxygen flow and fetal glucose delivery as needed to maintain fetal $PaO_2$ at 20 mmHg and glucose at 1.0 mmol/l for 10 days as reported previously (Camacho et al., 2022). For the FAS treatment ($n = 8$ fetuses: 3 males, 5 females) a maternal tracheal insufflation of humidified air and intravenous sterile saline into the fetus was given at fixed rates of 2 l/min for air and 1 ml/h for saline. The age-matched control fetuses had fetal catheters placed but received no chronic infusions (Fig. 1). Fetal arterial blood samples were collected every morning from day 0, before the intervention started (pre-intervention), until day 10, which was the final day of the intervention and day of postmortem tissue collection. Plasma samples were also collected for hormone and amino acid measurements throughout the treatment period as previously reported (Camacho et al., 2022). Growth device measurements of one-half thoracic circumference were taken at approximately 08.00 h and 14.00 h each day, and these measurements were averaged to obtain the daily measurement used to determine linear growth rates (Rumball et al., 2009). GSIS was measured on the eighth day of the intervention. Tissues were collected postmortem on day 10 as described previously (Camacho et al., 2022).

## Glucose and arginine stimulated insulin secretion

A square-wave hyperglycaemic clamp was performed to determine maximum glucose-stimulated insulin concentrations as previously described (Camacho et al., 2022; Green et al., 2011). Within 5 min after the final hyperglycaemic blood sample was collected, a bolus of arginine (0.5 mmol/kg estimated fetal weight) was infused over 4 min to determine GPAIS. Three arterial blood

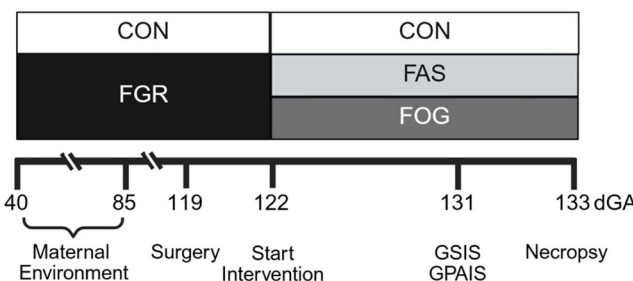

**Figure 1. Experimental design, groups and intervention conditions**
A schematic of the study design is shown to represent the two experimental groups (control [CON] and fetal growth restricted [FGR]) over the experimental timeline in days of gestational age (dGA). To create placental insufficiency and FGR pregnant ewes were maintained in a worm ambient environment for ~45 days (40–85 dGA), whereas ewes in the CON group were maintained in a thermoneutral environment. Following surgery on 119 ± 2 dGA, FGR fetuses were divided randomly into two perinatal interventions (i.e. experimental conditions): those that were supplemented with oxygen and glucose (FOG) and those that were supplemented with air and saline (FAS). The experimental treatment began on day 122 ± 2 and continued until day 133 ± 2 of gestation when the ewes and fetuses were humanly killed for postmortem tissue collection. On the 8th day of the perinatal intervention insulin responsiveness was measured during a glucose-stimulated insulin secretion (GSIS) challenge with a follow-on glucose-potentiated arginine-stimulated insulin secretion (GPAIS) test.

samples were collected at 5-min intervals immediately after the bolus was initiated.

### Biochemical analysis

Blood gases and oximetry were measured with an ABL 825 (Radiometer, Copenhagen, Denmark), and plasma glucose concentrations were measured with a YSI model 2900 SELECT Biochemistry Analyzer (Yellow Springs Instruments, Yellow Springs, OH, USA). Plasma insulin, cortisol, norepinephrine and IGF-1 were measured with enzyme-linked immunosorbent assays (ELISAs) as reported previously (Camacho et al., 2022; White et al., 2025). Arterial amino acid concentrations were measured using a Dionex 300 model 4500 amino acid analyser (Dionex, Sunnyvale, CA, USA) after deproteinization with sulfosalicyclic acid (Stremming et al., 2024).

### Histological analysis

Portions of the pancreas and semitendinosus muscle were fixed, embedded in O.C.T. Compound (Tissue-Tek), and tissue sections were affixed to slides as previously described (Cole et al., 2009; Yates et al., 2014). Sections were rinsed in phosphate buffered saline (PBS) with 0.1% Triton X-100, followed by citric acid antigen retrieval procedures (Cole et al., 2009). Non-specific binding was blocked with a NEN blocking reagent (Revvity, Waltham, MA, USA). For the pancreas primary antibodies used to identify $\beta$-cell proliferation were guinea pig anti-porcine insulin (1:500; Agilent Cat# A0564, RRID:AB_10013624) and rabbit polyclonal anti-phospho-Histone H3 (pHH3, 1 μg/ml; Millipore Cat# 06–570, RRID:AB_310177). For the semitendinosus muscle anti-mouse MHCI antibodies (MHCI; 1:25; DHSB; RRID: AB_2235587) were used to identify type I fibres with anti-rabbit laminin (laminin; 1:1000; Abcam; RRID: ab11575) used to identify the basal lamina. Primary antibodies against anti-rabbit Ki67 (Ki67; 1:1000; Abcam; RRID: ab15580) and anti-mouse pax7 (pax7; 1:20; DHSB; RRID: AB_528428) were used to measure satellite cell proliferation. Following an overnight incubation at 4°C, complexes were detected with anti-guinea pig, rabbit or mouse IgG antibodies conjugated to Alexa-488 or Alexa-594 (1:2000; Jackson Immuno Research Laboratories), and nuclei were counterstained with DAPI (4′,6 diamidino-2-phenylindole; Vector Laboratories, Burlingame, CA, USA). Tissue sections were rinsed in water, coverslips were mounted with 50% glycerol in 10 mM Tris–HCl (pH 8.0) and fluorescent microscopic images were taken using a DM5500 Leica Microscope System with Hamatsu camera and HC Image Live Hamatsu software. New identification numbers were given to animals that were not associated with treatment conditions to avoid counting bias.

For the pancreas morphometric analysis was performed on $\geq 25$ fields of view per tissue section that were taken at random over the entire section for $\beta$-cell area measurement ($\geq 3$ sections/fetus separated by $\geq 100$ μm). Area measurements were calculated using the per cent insulin positive area divided by the total tissue area per field of view. $\beta$-cells (insulin+ and DAPI+) and proliferating $\beta$-cells (insulin+, pHH3+ and DAPI+) were counted ($\sim$13,000 $\beta$-cells/animal) to determine the percentage of proliferating $\beta$-cells.

Semitendinosus muscle fibre cross sections were determined with $\geq 10$ fields of view per animal, which resulted in $\geq 200$ type I positive fibres per animal. Cross-sectional area (CSA) measurements were performed with Image J software and a Cross Sectional Analyzer plugin. Satellite cells (Pax7+ and DAPI+; >1000 cells per animal) and proliferating satellite cells (Pax7+, Ki67+ and DAPI+; >50 per animal) were counted to determine the percentage of proliferating satellite cells.

### Statistical analysis

Daily measurements over time were compared using mixed analysis of variance (ANOVA) procedures with fetus as the random effect (SAS 9.4, SAS Institute Inc., Cary, NC, USA). Main effects of the model were experimental group (CON, FAS and FOG), day of experimental intervention and their interaction, and differences ($P < 0.05$) were separated with a *post hoc* Tukey–Kramer test. To assess the overall response to experimental intervention (condition) we compared baseline values (pre-intervention; day 0) to the average values during the intervention (average of days 1–10). The average of entire intervention period was chosen for our analysis because we have previously shown that chronic, rather than acute, supplementation strategies are required to improve glucose tolerance (Camacho et al., 2022; Davis et al., 2021). For this analysis mixed ANOVA procedures were used for the main effects of experimental group (CON, FAS and FOG), experimental condition (pre-intervention and intervention) and their interaction; fetus was included as the random effect and differences identified with a Tukey–Kramer test. IGF-1 and NE concentrations were measured on days 0 and 10 and were analysed for group (CON, FAS and FOG) and experimental condition (pre-intervention and intervention) effects. Plasma concentrations of amino acids were collected on days 0, 5, 7 and 10 analysed for group (CON, FAS and FOG) and experimental condition (pre-intervention and intervention) effects after averaging concentrations during the intervention (days 5–10). GSIS and GPASIS differences were analysed using the mixed-ANOVA ($P < 0.05$). Growth rate differences were

measured as the slope of daily growth and were analysed by one-way ANOVA ($P < 0.05$). Cell proliferation rates and areas were calculated as the percentage of proliferating cells or positive areas, respectively, and were also analysed using a one-way ANOVA. Simple linear regression models were made using GraphPad Prism (version 10). Fetal sex differences were not analysed due to the paucity of animals; therefore data from male and female fetuses were combined within experimental groups.

## Results

### Oxygen and glucose supplementation improves physiological parameters that support growth

On day 0 before starting the intervention, blood $PaO_2$ levels were not different between FGR groups and lower compared with CON fetuses (Fig. 2*A*). During the 10-day OG supplementation, the condition average $PaO_2$ increased ($P = 0.00390$) in FOG fetuses to levels that were not different from CON fetuses ($P = 0.118$). Daily blood $O_2$ contents were lower in FAS and FOG fetuses compared with CON fetuses and did not increase during the OG intervention (group × day interaction $P = 0.0754$), and this was also observed with the average for experimental condition (Fig. 2*B*). Fetal haematocrit and haemoglobin concentrations were not different between experimental groups in either experimental condition (Fig. 2*C* and *E*). However haematocrit and haemoglobin decreased ($P < 0.0001$) in FOG fetuses during the OG intervention, whereas these values remained stable in FAS and CON groups throughout the intervention period.

Blood pH values were not different between experimental groups on day 0 and remained unchanged in FAS and CON fetuses throughout the study. The group × day interaction ($P = 0.0292$) revealed a decline in FOG pH values between days 2–3 and days 7–10 (Fig. 2*D*). Although no interaction was found for group × condition pH averages, the average pH in the FOG fetus for the entire study period ($7.28 \pm 0.05$, $P = 0.0292$) was lower than the pH of the FAS fetus ($7.33 \pm 0.03$), but neither differed from CON pH values ($7.31 \pm 0.03$). Daily excess base concentrations were not different between experimental groups on day 0 but declined in FOG fetuses between day 3 and days 7–10 (Fig. 2*F*). No statistical differences in excess base concentrations were observed when experimental group and condition averages were compared.

At day 0 before the intervention, plasma glucose (FOG $P = 0.00810$ & FAS $P = 0.0005$) and insulin (FOG $P = 0.0205$ & FAS $P < 0.00130$) concentrations were lower in FGR groups compared with CON fetuses (Fig. 2*G* and *H*). Glucose and insulin concentrations were not different between FOG and FAS groups on day 0. Daily glucose concentrations remained lower in FAS fetuses

compared to CON fetuses throughout the intervention period except for days 2 and 4. In FOG group daily glucose concentrations were not different from CON fetuses throughout the intervention. Daily FOG glucose concentrations were greater than FAS fetuses on days 2, 3, 5, 6, 7, 8 and 10. During the intervention daily insulin concentrations were lower in FAS fetuses compared to CON fetuses for each day except on days 2 and 4. Daily FOG insulin concentrations were not different from FAS and CON concentrations during the intervention. For the experimental condition averages (pre *vs*. intervention), glucose concentrations increased in FOG fetuses ($P < 0.0001$), and the intervention average was not different ($P = 0.999$) from CON fetuses, whereas average glucose concentrations during the FAS intervention period were lower than both CON ($P = 0.0002$) and FOG ($P = 0.0007$) averages. No group × condition interaction ($P = 0.106$) was found for insulin, and insulin concentrations were lower in FOG ($0.26 \pm 0.15$ µg/l, $P = 0.0083$) and FAS ($0.11 \pm 0.12$ µg/l, $P < 0.0001$) compared with CON fetuses ($0.50 \pm 0.23$ µg/l).

On day 0 IGF-1 concentrations were not different between FGR groups ($P = 0.740$), but IGF-1 concentrations in FGR groups were lower than that in the CON group (FOG $P = 0.0002$ & FAS $P < 0.0001$; Fig. 3). During the intervention IGF-1 concentrations increased ($P = 0.0499$) in FOG fetuses but remained lower ($P = 0.0060$) than CON IGF-1 concentrations.

For norepinephrine no group × condition interaction was observed ($P = 0.651$), but there were group differences ($P < 0.0001$). Norepinephrine concentrations were greater in FAS ($2383 \pm 2092$ pg/ml, $P < 0.0001$) and FOG ($936 \pm 958$ pg/ml, $P = 0.00410$) fetuses compared with CON fetuses ($290 \pm 210$ pg/ml), regardless of condition.

Fetal plasma cortisol concentrations were not different between groups before (FAS $11.6 \pm 3.3$, FOG $8.6 \pm 2.6$, CON $5.9 \pm 0.9$ ng/ml) or during the intervention (FAS $14.9 \pm 4.6$, FOG $9.3 \pm 2.7$, CON $5.7 \pm 0.5$ ng/ml; group $P = 0.0616$, group × condition $P = 0.665$).

Fetal plasma amino acid concentrations were analysed by experimental condition means (Tables 1 and 2). Significant group × condition interactions were found for three essential amino acids: leucine, methionine and phenylalanine. However the Tukey–Kramer test only resolved condition differences (reductions) in leucine and phenylalanine concentrations but not methionine concentrations (Table 1). Leucine and phenylalanine decreased in all groups and were not different from CON concentration at the respective condition. Group differences without an interaction with experimental condition were found for isoleucine, lysine, threonine and tryptophan (Table 2). Isoleucine concentrations were lower in FOG fetuses compared with CON fetuses ($P = 0.0104$). In FAS fetuses isoleucine concentrations were not different from those in FOG ($P = 0.547$) and

**Table 1. Fetal arterial amino acid concentrations with group × condition interactions.**

| Group | CON (n = 9) | | FAS (n = 8) | | FOG (n = 8) | | ANOVA statistics | | |
|---|---|---|---|---|---|---|---|---|---|
| Condition | Pre | Int | Pre | Int | Pre | Int | Group | Condition | G × C |
| **Essential (µmol/l)** | | | | | | | | | |
| Leucine | 244 ± 59[a] | 194 ± 54[bc] | 262 ± 69[a] | 159 ± 37[c] | 228 ± 51[ab] | 135 ± 31[c] | 0.126 | <0.0001 | 0.0411 |
| Phenylalanine | 139 ± 45[abc] | 119 ± 26[abcd] | 144 ± 58[a] | 97 ± 21[bc] | 142 ± 23[ab] | 91 ± 25[d] | 0.573 | <0.0001 | 0.0390 |
| **Non-essential (µmol/l)** | | | | | | | | | |
| Alanine | 249 ± 55 | 286 ± 58 | 373 ± 114 | 236 ± 55[#] | 324 ± 54 | 298 ± 91 | 0.250 | 0.0036 | <0.0001 |
| Asparagine | 35 ± 7[b] | 44 ± 11[ab] | 53 ± 11[a] | 49 ± 12[ab] | 50 ± 9[ab] | 46 ± 11[ab] | 0.0223 | 0.978 | 0.0276 |
| Glutamine | 496 ± 106 | 501 ± 73 | 492 ± 77 | 397 ± 78[#] | 522 ± 58 | 440 ± 105[#] | 0.355 | <0.0001 | 0.0034 |

*Note*: Individual amino acid concentrations were analysed for group (CON, FAS and FOG), experimental condition and the group × condition interaction (G × C). Experimental conditions are pre-intervention (day 0), concentrations (Pre) and intervention (Int), which is the average value for days 5, 7 and 10. Data were analysed with a mixed model ANOVA using fixed effects for experimental group, condition and their interaction (*F* test) and fetus as the random effect. Mean (±SD) is presented for amino acids with significant interactions. Mean differences determined with a Tukey–Kramer test are identified with different letters. Based on the Tukey–Kramer test the # symbol represents differences between experimental conditions within the group, but all other pairwise comparisons were not significant.

**Table 2. Fetal arterial amino acid concentrations without group × condition interactions.**

| Group | CON (n = 9) | FAS (n = 8) | FOG (n = 8) | Group | Condition | G × C |
|---|---|---|---|---|---|---|
| **Essential (µmol/l)** | | | | | | |
| Histidine | 58 ± 14 | 55 ± 10 | 62 ± 18 | 0.430 | 0.0921 | 0.106 |
| Isoleucine | 120 ± 32[a] | 91 ± 34[ab] | 80 ± 23[b] | 0.0120 | <0.0001 | 0.0592 |
| Lysine | 96 ± 26 | 89 ± 35 | 70 ± 23 | 0.0360 | 0.280 | 0.0545 |
| Methionine | 86 ± 20 | 73 ± 14 | 77 ± 15 | 0.437 | 0.0281 | 0.0138 |
| Threonine | 358 ± 110[a] | 280 ± 66[ab] | 257 ± 87[b] | 0.0270 | <0.0001 | 0.614 |
| Tryptophan | 53 ± 13[a] | 38 ± 9[b] | 40 ± 11[b] | 0.0151 | 0.0139 | 0.323 |
| Valine | 607 ± 186 | 526 ± 197 | 475 ± 172 | 0.292 | <0.0001 | 0.0884 |
| **Non-essential (µmol/l)** | | | | | | |
| Arginine | 105 ± 25[a] | 65 ± 20[b] | 69 ± 22[b] | 0.0014 | 0.273 | 0.294 |
| Aspartate | 25 ± 5 | 27 ± 7 | 27 ± 7 | 0.780 | 0.0174 | 0.403 |
| Cysteine | 18 ± 6 | 18 ± 7 | 21 ± 14 | 0.499 | 0.521 | 0.264 |
| Glutamate | 45 ± 14 | 42 ± 17 | 44 ± 17 | 0.645 | <0.0001 | 0.101 |
| Glycine | 389 ± 75 | 394 ± 134 | 476 ± 135 | 0.161 | 0.0197 | 0.521 |
| Ornithine | 75 ± 26 | 81 ± 26 | 58 ± 31 | 0.288 | 0.240 | 0.189 |
| Proline | 194 ± 77 | 232 ± 143 | 229 ± 164 | 0.437 | 0.230 | 0.040 |
| Serine | 820 ± 273[a] | 495 ± 200[b] | 791 ± 308[ab] | 0.0334 | <0.0001 | 0.121 |
| Taurine | 101 ± 58 | 221 ± 132 | 149 ± 115 | 0.124 | 0.510 | 0.0119 |
| Tyrosine | 135 ± 31[a] | 115 ± 38[ab] | 90 ± 18[b] | 0.0260 | 0.0765 | 0.131 |

*Note*: Individual amino acid concentrations were analysed for group (CON, FAS and FOG), experimental condition and the group × condition interaction (G × C). Experimental conditions are pre-intervention (day 0), concentrations (Pre) and intervention (Int), which is the average value for days 5, 7 and 10. Data were analysed with a mixed model ANOVA using fixed effects for experimental group, condition and their interaction (*F* test) and fetus as the random effect. Experimental group means (±SD) are presented for amino acids without a group × condition interaction. Mean differences were determined with a Tukey–Kramer test and are identified with different letters. Although the *P*-value associated with the F-statistic for the interaction was significant for methionine, proline and taurine, no differences between means were detected with the Tukey–Kramer test. Similarly no group differences were identified with the *post hoc* test for lysine.

CON ($P = .107$) groups. Lysine concentrations were not different in FOG fetuses compared with either FAS ($P = 0.0579$) or CON ($P = 0.0619$) fetuses with the *post hoc* test. Compared with CON fetuses threonine concentrations were lower in FOG fetuses ($P = 0.0358$), and FAS threonine concentrations were not different between FOG ($P = 0.9342$) and CON ($P = 0.0759$)

groups. Tryptophan concentrations were lower in FAS ($P = 0.0220$) and FOG ($P = 0.0458$) fetuses compared with CON fetuses.

Significant group × condition interactions were found for non-essential amino acids alanine, asparagine, glutamine, proline and taurine, although Kramer–Tukey test only identified condition differences for alanine (FAS

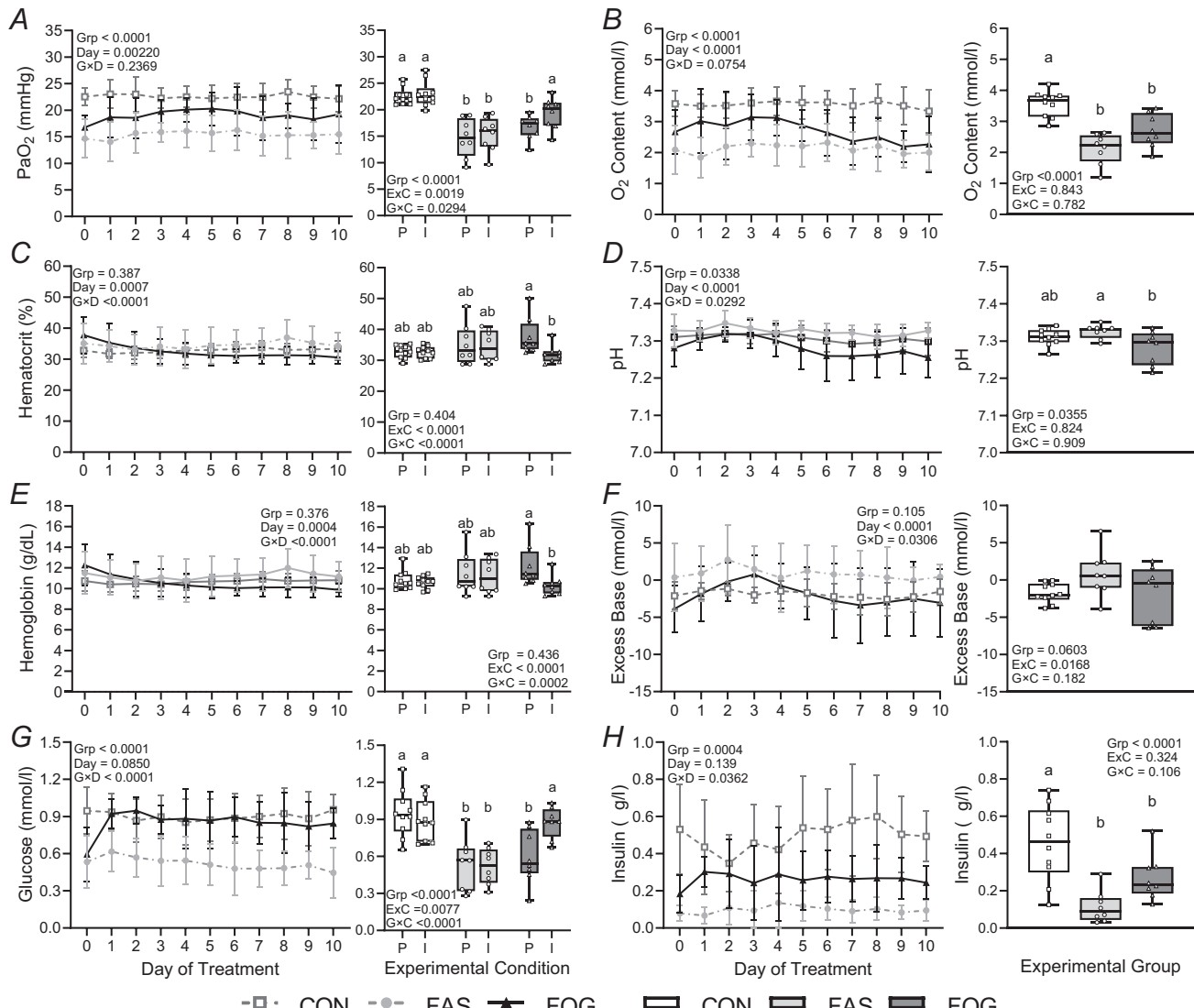

**Figure 2. Oxygen and glucose supplementation increased PaO₂ and glucose concentrations**
After sample collection on day 0 administration of oxygen + glucose or air + saline (intervention) was initiated in fetuses with FGR. The maternal insufflation of oxygen/air and fetal intravenous infusion of glucose/saline was maintained for 10 days. For the line graphs the *x*-axis is the day of treatment, and the mean (±SD) values are presented for FAS (*n* = 8; FGR fetuses administered air and saline), FOG (*n* = 8; FGR fetuses administered oxygen and glucose) and CON (*n* = 10; no intervention other than daily sampling). Daily averages are presented for the partial pressure of oxygen (PaO₂, panel *A*), blood oxygen content (O₂, panel *B*), fetal haematocrit (panel *C*), fetal pH (panel *D*), haemoglobin (panel *E*), excess base (panel *F*), plasma glucose (panel *G*) and plasma insulin (panel *H*). Box and whisker plots with individual fetal means summarize the two experimental conditions: preintervention (day 0 measurements = P) and intervention (average of days 1–10 = I) values. Differences (*P* < 0.05) were determined with a mixed-model ANOVA and *post hoc* Tukey–Kramer test and identified by different letters in summary graphs on the right. *P*-values for the analysis are presented for group (G), day (D), condition (C) and their interaction (G × D or G × C).

group) and glutamine (FAS and FOG groups; Table 1). Pre-intervention asparagine concentrations were greater ($P = 0.0335$) in FAS fetuses compared with CON fetuses, and FOG fetuses were not different from FAS ($P = 0.996$) or CON ($P = 0.0937$) fetuses. Asparagine averages during the intervention were similar among groups. Group differences without condition interactions were found for non-essential amino acids: arginine, serine and tyrosine (Table 2). Compared with CON fetus arginine concentrations were lower in FOG ($P = 0.00500$) and FAS ($P = 0.00310$). Serine concentrations were lower in FAS fetuses ($P = 0.0495$) compared with CON, whereas FOG serine concentrations were not different from FAS ($P = 0.0643$) or CON ($P = 0.998$) fetuses. Tyrosine concentrations were lower in FOG fetuses ($P = 0.0213$) compared with CON fetuses, whereas FAS fetal tyrosine concentrations were not different from those in FOG ($P = 0.171$) and CON ($P = 0.594$) fetuses.

### Fetal intervention improves glucose but not arginine-stimulated insulin secretion

On day 8 of the intervention GSIS and GPAIS were tested. During the GSIS study baseline glucose concentrations were lower in FAS fetuses compared to FOG ($P = 0.00730$) and CON ($P = 0.0005$) fetuses, which were not different ($P = 0.910$; Fig. 4A). During the hyperglycaemic period glucose concentrations increased

($P < 0.0001$) from baseline and were not different between experimental groups. At baseline, insulin concentrations were not different between groups. Glucose-stimulated insulin concentrations increase (CON $P < 0.0001$; FAS $P = 0.00820$; FOG $P < 0.0001$) from baseline concentrations in all experimental groups (Fig. 4B). Glucose-stimulated insulin concentrations were greater in FOG ($P = 0.00150$) and CON ($P = 0.00630$) fetuses compared with FAS fetuses, and maximal glucose-stimulated insulin concentrations were not different between FOG and CON fetuses ($P = 0.981$; Fig. 4B). A positive linear relationship was predicted for maximal glucose-stimulated insulin concentrations and basal $PaO_2$ values in FAS fetuses (slope 0.191, $R^2$ 0.708, $P = 0.00872$); however the slopes for FOG fetuses (slope 0.139, $R^2$ 0.332, $P = 0.0638$) and CON fetuses (slope $-0.236$, $R^2$ 0.366, $P = 0.135$) were not different from zero (Fig. 4C). The GPAIS analysis showed that there was an interaction ($P < 0.0001$) between experimental group and time (minute, Fig. 4D). In FAS and FOG groups glucose-potentiated arginine-stimulated insulin concentrations were lower than CON insulin concentrations at 5 (FAS $P \leq 0.0001$, FOG $P = 0.0247$) and 10 min (FAS $P = 0.0004$, FOG $P = 0.0201$). Insulin concentrations following the administration of arginine were not different between FOG and FAS fetuses.

### Oxygen and glucose supplementation increased linear growth rates in FGR fetuses

Fetal thoracic circumference (girth) increased in all experimental groups during the intervention period (Fig. 5). During the intervention linear growth rates in FOG fetal girth ($1.8 \pm 0.4$ mm/day) were greater ($P = 0.000509$) than FAS fetal girth growth rates ($0.9 \pm 0.3$ mm/day). Linear growth rates in CON fetal girth ($2.1 \pm 0.3$ mm/day) were similar to FOG fetuses ($P = 0.336$) and greater ($P < 0.0001$) than FAS fetuses. The initial half-length of their girth-measuring device was not different from FAS fetuses (FOG $= 116.5 \pm 2.9$ mm *vs.* FAS $= 113.9 \pm 1.3$ mm, $P = 0.985$). In CON fetuses the initial half girth device length ($146.8 \pm 2.3$ mm) was greater than the lengths for FOG ($P = 0.00402$) and FAS ($P = 0.0277$) fetuses.

At the conclusion of the 10-day intervention, FAS and FOG fetal body weights were 54% and 45% less, respectively, when compared with CON fetuses despite having similar gestational ages (Table 3). Fetal organ and muscle weights were all lighter in FAS and FOG fetuses compared with CON fetuses. Placentome mass was lower for FAS and FOG fetuses compared with CON fetuses, which resulted in greater placental efficiency. Brain weight relative to body weight was greater for FAS and FOG fetuses compared with CON fetuses. The relative weights

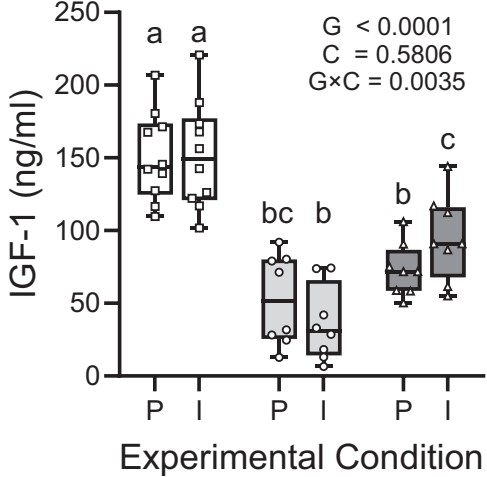

**Figure 3. IGF-1 concentrations increased with oxygen and glucose supplementation**
Plasma insulin-like growth factor 1 (IGF-1) concentrations were measured in CON ($n = 10$), FAS ($n = 8$) and FOG ($n = 8$) groups (G). Experimental condition (C) means of IGF-1 concentrations are presented for pre-intervention (day 0, P) and intervention (day 10, I) with a box and whisker plot that includes individual fetus means. Data were analysed with mixed-model ANOVA procedures. Differences ($P < 0.05$) were determined with a *post hoc* Tukey–Kramer test and identified by different letters.

for liver, heart, lungs and kidneys were not different between experimental groups. The relative weights for biceps femoris and semitendinosus muscles were lower in FAS fetuses than FOG and CON fetuses.

fetuses compared with FOG and CON fetuses, respectively (Fig. 6*A*). The mean pancreatic insulin+ area was not different ($P = 0.1138$) between experimental groups (Fig. 6*B*).

### Oxygen and glucose supplementation increased rates of β-cell proliferation

Rates of β-cell mitosis and the percentage of insulin positive (β-cell) area were determined for all experimental groups. The percentage of insulin+ pHH3+ cells was 56% ($P = 0.0103$) and 58% ($P = 0.00545$) lower in FAS

### Oxygen and glucose supplementation increased rates of satellite cell proliferation

Rates of satellite cell proliferation and type 1 myofibre CSA were measured in the semitendinosus muscle of the fetus. The percentage of proliferating satellite cells (pax7+ Ki67+) was 2.2-fold greater ($P = 0.0298$) in FOG

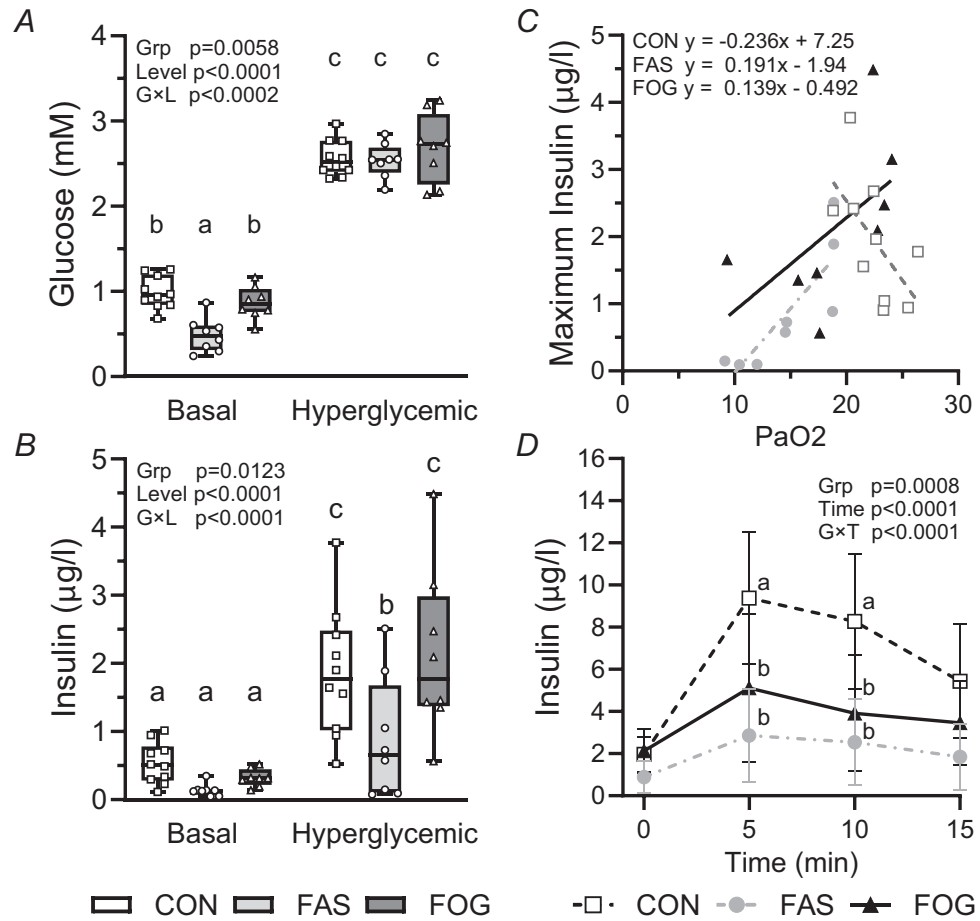

**Figure 4. Oxygen and glucose supplementation increased glucose-stimulated insulin concentrations**
Plasma glucose *A* and insulin *B* concentrations during the basal and hyperglycaemic periods of the glucose-stimulated insulin secretion (GSIS) study are presented for CON ($n = 10$), FAS ($n = 8$) and FOG ($n = 8$). The individual fetal measurements are represented by dots, and the data are summarized with box and whiskers plots for each experimental group. Maximum insulin concentrations for each fetus are presented relative to the basal fetal PaO$_2$ levels *C*, and regression lines are shown for each group (CON $R^2$ 0.366, FAS $R^2$ 0.708 and FOG $R^2$ 0.332). In panel *D* the follow-on glucose-potentiated arginine-stimulated insulin secretion (GPAIS) means (±95% confidence interval [CI]) are presented. Comparisons were made with a two-way ANOVA, with experimental group and glucose level for GSIS and group and time for GPAIS. Differences ($P < 0.05$) between GSIS group × level or GPAIS group × time means were identified with a Tukey–Kramer test and indicated by different letters. Linear regression analysis was used to determine slope difference between groups ($P = 0.0172$) in panel *C*. The FAS slope was different from zero ($P = 0.00872$; 95% CI 0.069 to 0.314), whereas slopes for CON ($P = 0.0638$; 95% CI −0.489 to 0.0172) and FOG ($P = 0.135$; 95% CI −0.0579 to 0.335) were not different from zero.

**Table 3. Postmortem age and weights for the fetus, placenta and fetal organs.**

| Parameters | CON (10) | FAS (8) | FOG (8) | *P*-value |
|---|---|---|---|---|
| Gestation age (d) | 132 ± 2 | 134 ± 3 | 132 ± 1 | 0.0913 |
| Fetal weight (g) | 3438 ± 491[a] | 1579 ± 485[b] | 1884 ± 443[b] | <0.0001 |
| Placental weight (g) | 409 ± 100[a] | 135 ± 61[b] | 155.1 ± 74[b] | <0.0001 |
| Placentome # | 78 ± 13 | 61 ± 18 | 60 ± 18 | 0.0459 |
| Placental efficiency | 8.7 ± 1.6[a] | 12.4 ± 2.7[b] | 13.8 ± 4.7[b] | 0.00570 |
| Brain (g) | 49.9 ± 2.9[a] | 41.5 ± 7.3[b] | 40.8 ± 3.5[b] | 0.0006 |
| Liver (g) | 104.0 ± 27.0[a] | 47.0 ± 17.8[b] | 53.7 ± 14.6[b] | <0.0001 |
| Heart (g) | 24.8 ± 4.7[a] | 11.4 ± 3.9[b] | 14.8 ± 1.9[b] | <0.0001 |
| Lungs (g) | 108.9 ± 24.4[a] | 51.4 ± 16.8[b] | 57.8 ± 17.1[b] | <0.0001 |
| Kidneys (g) | 20.7 ± 3.9[a] | 9.6 ± 2.2[b] | 12.7 ± 3.1[b] | <0.0001 |
| Biceps femoris (g) | 37.5 ± 5.9[a] | 13.8 ± 4.8[b] | 19.3 ± 4.7[b] | <0.0001 |
| Semitendinosus (g) | 12.8 ± 2.2[a] | 5.1 ± 2.2[b] | 7.0 ± 2.0[b] | <0.0001 |
| Relative brain (g/kg) | 14.8 ± 2.1[a] | 27.5 ± 5.3[b] | 22.6 ± 4.9[b] | <0.0001 |
| Relative liver (g/kg) | 30.0 ± 4.8 | 29.3 ± 3.5 | 28.4 ± 2.7 | 0.681 |
| Relative heart (g/kg) | 7.2 ± 0.8 | 7.2 ± 1.0 | 8.1 ± 1.4 | 0.167 |
| Relative lungs (g/kg) | 31.5 ± 3.8 | 32.6 ± 4.4 | 30.2 ± 4.4 | 0.532 |
| Relative kidney (g/kg) | 6.0 ± 0.6 | 6.4 ± 1.4 | 6.8 ± 0.7 | 0.258 |
| Relative biceps femoris (g/kg) | 10.9 ± 1.0[a] | 8.6 ± 1.1[b] | 10.3 ± 1.4[a] | 0.00140 |
| Relative semitendinosus (g/kg) | 3.7 ± 0.3[a] | 3.1 ± 0.5[b] | 3.7 ± 0.6[a] | 0.0180 |

*Note*: Data are presented as mean ± standard deviation and analysed by one-way ANOVA. Animal numbers within groups are defined in parentheses. Group differences ($P < 0.05$) were determined with a Tukey–Kramer test, and the different letters indicate differences between groups. Relative weights (g/kg) are calculated as the organ weight divided by body weight.

muscle compared with FAS muscle (Fig. 7*A*). Compared to CON fetuses, FOG and FAS satellite cell proliferation rates were 45% ($P = 0.00143$) and 75% ($P < 0.00001$) lower, respectively. Type I myofibre mean CSA was smaller in FAS fetuses compared with CON type I myofibres ($P = 0.00818$, Fig. 7*B*). Type I myofibre CSA for FOG semitendinosus muscle was not different from either FAS ($P = 0.655$) or CON ($P = 0.0562$) muscles.

## Discussion

At present no therapeutic interventions exist in practice for fetuses experiencing growth restriction due to placental insufficiency outside of timed delivery (Davenport et al., 2022; Lausman et al., 2013). We have devised and tested an intervention that simultaneously supplements oxygen and glucose to fetuses with established FGR because treatment initiation requires clinical diagnosis. Additionally placental transport of oxygen and glucose is regulated in part by their diffusion gradients, and therefore increasing maternal concentrations increases fetal concentrations and negates direct fetal access, although in this study glucose was administered to the fetus intravenously. Previously we have shown that chronic supplementation of glucose for 5 days improves fetal insulin secretion and action in FGR fetuses without increasing fetal body weights (Camacho

et al., 2022). In this study we have extended the duration of the supplementation and show faster rates of linear growth in FGR fetuses, which are supported by greater proliferation rates for pancreatic $\beta$-cells and skeletal muscle satellite cells, two tissues that regulate glucose homeostasis. Findings from these studies demonstrate that the combined delivery of oxygen and glucose normalizes growth in fetuses with established placenta insufficiency and FGR. Consequently this approach offers a feasible clinical therapy for managing FGR secondary to placental insufficiency.

Our prenatal intervention increased $PaO_2$ and glucose concentrations, which partially corrected IGF-1 concentrations, and together increased fetal thoracic linear growth rates and proliferation rates of skeletal muscle satellite cells. Of note relative biceps femoris and semitendinosus muscle weights (normalized to body weight) were restored to control levels in FOG fetuses, indicating partial rescue of skeletal muscle mass. Importantly fetal amino acid concentrations were not markedly reduced during the oxygen and glucose intervention. These findings support the premise that substrate availability for oxidative metabolism is fulfilled with supplemented glucose, which is predicted to redirect amino acid utilization away from oxidative metabolism to protein accretion (Ross et al., 1996; Wai et al., 2018). Additionally GSIS normalized with the longer intervention whereas GPAIS did not improve. These findings

indicate that the restoration of insulin secretion and β-cell proliferation rates may not include normalization of pancreas or islet insulin content (Camacho et al., 2022; Limesand et al., 2006). Interestingly we found GSIS was dependent on $PaO_2$ in FAS fetuses. This negative

association identifies a critical threshold for oxygen tension at ~18 mmHg because oxygen did not limit insulin secretion in FOG and CON groups. Together the current set of experiments shows that the combined supplementation of oxygen and glucose rescues several critical limitations associated with FGR.

Our primary objective was to show that oxygen and glucose replacement restored fetal linear growth rates, despite not seeing meaningful increases in absolute body weight. In the FAS fetus the average thoracic linear growth rate was 42% of the growth rate in control fetuses. This reduction matches earlier findings with hind limb measurement showing slower linear growth in this model of FGR (Rozance et al., 2018). Importantly we show an immediate increase in fetal thoracic growth from the onset of the intervention that parallels control fetuses. Because the initial girth length was significantly shorter in FGR groups, the normalized growth trajectory in FOG fetuses will most likely require an earlier start, more time or optimally both parameters to increase body mass without catch-up growth. Increases in faltering growth rates are consistent with previous reports. One study demonstrated that linear growth rates were restored by long-term (30 day) IGF-1 administration in fetuses that were previously growth restricted with utero-placental embolization (Eremia et al., 2007; Wali et al., 2012). Other reports show that refeeding after short-term severe maternal starvation (≤16 days) rescued fetal growth (Mellor & Matheson, 1979; Mellor & Murray, 1982). However there is also evidence for persistent stunting in fetal growth, which occurred after severe long-term maternal starvation (21 days) (Mellor & Murray, 1982) or after only 1 week of IGF-1 administration in near-term fetuses in this FGR model (White et al., 2025). Therefore

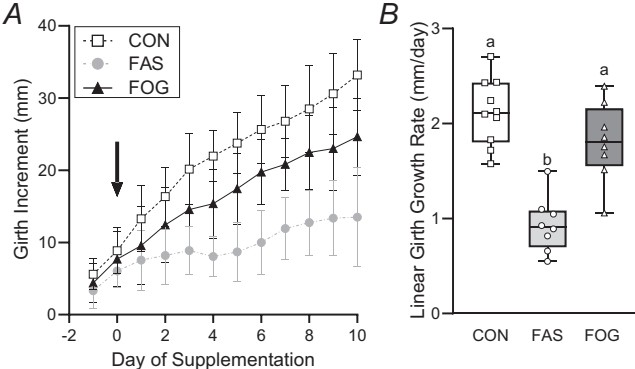

**Figure 5. Fetal supplementation increased linear growth rates of fetal growth restriction (FGR) fetuses**
The incremental change in length for one half of the fetal thoracic circumference (girth) is presented relative to the day of supplementation. Incremental increases in the fetal girth length were calculated retrospectively from their initial length measurement and expressed as the mm accumulated increment of 50% of the fetal girth (sternum midline to spinal process). Average linear growth measurements (±95% CI) are presented from day −1 and continued throughout the intervention that began on day 0 (arrow). Panel *A* shows the incremental growth for FAS (*n* = 8), FOG (*n* = 8) and CON (*n* = 10) fetuses. Panel *B* shows the box plots of fetal growth rates (mm/day) in each experimental group, with dots representing growth rates of the individual fetuses. Linear growth rates were analysed with a one-way ANOVA (*P* < 0.0001 for group differences). Differences between experimental groups (CON, FAS and FOG) were determined using the Tukey–Kramer test (*P* ≤ 0.0002) and are indicated by different letters.

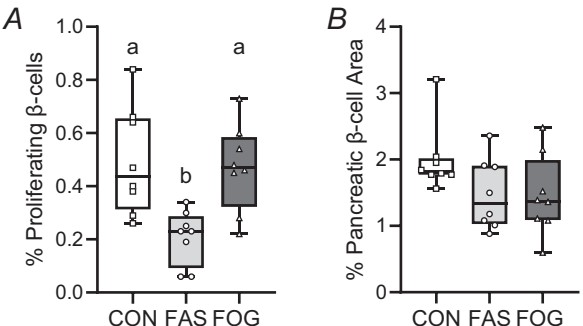

**Figure 6. Fetal treatment increased rates of β-cell proliferation**
β-cell proliferation rates *A* and per cent β-cell area *B* are presented for CON, FAS and FOG fetuses (*n* = 8/group). Per cent proliferation was determined by the number of phosphorylated histone H3 (PHH3) positive insulin positive cells (β-cells) per total insulin positive β-cell. Insulin positive area was evaluated relative to total pancreas area. Individual animal means are shown with dots. Differences (*P* ≤ 0.01) were determined by ANOVA and Tukey–Kramer *post hoc* test and are represented by different letters.

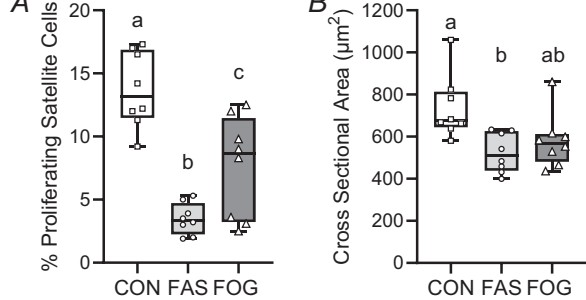

**Figure 7. Fetal treatment increased rates of satellite cell proliferation**
Semitendinosus muscle was immunostained for Pax7 and Ki67 to identify proliferating satellite cells (% Ki67+ Pax7+ per total Pax7+ cells, panel *A*). Cross-sectional area (μm²) for type 1 myofibres was measured after immunostaining semitendinosus muscle for MHCI and laminin (*B*). Box and whisker plots with individual fetal means are presented for FAS (*n* = 8), FOG (*n* = 8) and CON (*n* = 8) groups. Differences (*P* ≤ 0.05) were determined by ANOVA and Tukey–Kramer test and are represented by different letters.

despite differences in the induction of FGR, timing of the treatment remains an important aspect for the future translation of the intrauterine intervention.

Further support for the growth-promoting benefits of oxygen and glucose intervention was the increased proliferation rates of fetal pancreatic $\beta$-cells and skeletal muscle satellite cells. We focused on the proliferation rates of these tissues for two reasons. First both tissues are known targets for fetal programming of glucose intolerance, and second, proliferation rates of these cells are lower in FGR fetuses, highlighting a targetable mechanism for reestablishing growth (Brown, 2014; Davis et al., 2015; Limesand & Rozance, 2017; Limesand et al., 2005; Stremming et al., 2024; Yates et al., 2014). Despite restoring $\beta$-cell proliferation to normal rates in FOG fetuses, there were no differences in the percentage of $\beta$-cell area between groups. Although pancreases were not weighed in this study, we have shown previously that the relative proportion of pancreas weight to fetal body weight is maintained between FGR and control groups (Davis et al., 2015; Limesand et al., 2005). Therefore based on the relative decline in estimated pancreas weights we predict that $\beta$-cell mass is lower in FGR fetuses and that higher rates of $\beta$-cell proliferation will also reflect global increases in pancreas growth (Davis et al., 2015). Faster $\beta$-cell proliferation rates, along with normalized GSIS responsiveness, indicate that raising $PaO_2$ and glucose concentrations alleviate the inhibition of both insulin secretion and $\beta$-cell expansion in FGR fetuses (Kale et al., 2025; Limesand et al., 2005; Tsuyama et al., 2023; Wang et al., 2020).

Similar mechanisms are postulated for satellite cells because previous research has shown that hypoxic environments ($<5\%$ $O_2$) inhibit myoblast differentiation leading to atrophy (Cirillo et al., 2017; Sakushima et al., 2020; Zhao et al., 2025). Additionally fetal oxygen and glucose supplementation increased circulating IGF-1, which plays a key role in promoting muscle growth (Aguirre et al., 2016; Chang et al., 2021; Lin et al., 2021; Rozance et al., 2018; Stremming et al., 2021). Therefore a proposed mechanism is that IGF-1 activates growth promoting factors such as calcineurin signalling, phosphoinositol kinase and calmodulin kinase to activate quiescent satellite cells, initiating proliferation (Miretti et al., 2024; Romagnoli et al., 2021). When satellite cells are activated, cell division is triggered and myogenic progenitor cells differentiate to form myoblasts (Forcina et al., 2019). Myoblasts undergo fusion into myotubes promoting myofibre hypertrophy. The evidence presented from two distinct tissue types shows that the prenatal oxygen and glucose treatment promotes cell proliferation to support fetal growth.

The oxygen and glucose intervention lowered the fetal haematocrit, which paralleled haemoglobin concentrations. Despite the decrease, fetal haematocrit did not fall out of the normal range because concentrations were not different among experimental groups. These results indicate that reoxygenation of the FGR fetus may decrease erythropoietin concentration, thus lowering their haematocrit (Fisher, 2003; Kitanaka et al., 1989; Lim et al., 1996). The outcome from the decreased fetal haematocrit is that oxygen content in the FOG fetuses remained low and was not different from FAS fetuses, even though the $PaO_2$ was elevated.

Fetal oxygen and glucose supplementation had a minimal impact on arterial amino acid concentrations in FOG fetuses compared with FAS, except for phenylalanine which was not different between CON and FOG groups. No effect on amino acid concentrations despite proven reductions in the transplacental flux of essential amino acids in FGR fetal sheep indicates adaptations in fetal amino acid utilization (Anderson et al., 1997; Brown et al., 2021; Ross et al., 1996). In FGR fetuses lower rates of amino acid utilization were demonstrated with lower rates of leucine incorporation into proteins and lower fractional synthetic rates of skeletal muscle (Brown et al., 2012; Rozance et al., 2018). Reductions in protein synthesis and accretion represent a primary mechanism for FGR fetuses to conserve oxygen at the expense of growth, while also allowing amino acid to be a viable source for oxidative metabolism (Kennaugh et al., 1987; Rozance et al., 2018). In the FGR fetus lower amino acid utilization rates are dependent on fetal oxygenation (Regnault et al., 2013). Therefore a possible explanation for minimal differences in fetal arterial amino acid concentrations, despite normal rates of oxidative metabolism per unit of fetal weight, is that amino acid utilization returns to accretion and is no longer required for oxidative metabolism because of greater oxygen and glucose availability.

Alanine concentrations were maintained during oxygen and glucose supplementation, unlike the reductions observed in FAS fetuses. In FGR fetuses alanine is released from skeletal muscle, which may serve as a major substrate for prematurely active hepatic gluconeogenesis (Brown et al., 2024; Chang et al., 2019). The preservation of circulating alanine during the treatment indicates improved substrate availability and reduced reliance on muscle catabolism, thereby conserving amino acids for anabolic rather than catabolic pathways. A limitation for this study was that placental and fetal amino acid fluxes and protein accretion rates were not measured in FOG fetuses. However the results indicate that arterial amino acids are not limited during fetal treatment. Therefore we speculate that oxygen supplementation is sufficient to increase amino acid clearance rates and glucose supplementation is sufficient to fulfill the energetic requirements to increase protein accretion.

Glucose-stimulated insulin concentrations normalized following 8 days of glucose and oxygen supplementation, but glucose-potentiated arginine-stimulated insulin concentrations remained blunted. Difference between the two experiments indicates normalization of insulin secretion responsiveness but not islet insulin content, which may reflect deficiencies in insulin synthesis or $\beta$-cell mass. In previous reports that evaluated the actions of higher norepinephrine concentrations we propose that adrenergic receptors are desensitized and fetal $\beta$-cells have enhanced insulin secretion responsiveness due to tighter metabolic coupling with lower uncoupling protein 2 expression (Chen et al., 2014, 2017; Davis et al., 2021; Kelly et al., 2017). This concept of adrenergic receptor desensitization resulting in insulin hyper-secretion supports the improvement in insulin secretion with no reductions in norepinephrine concentrations (Limesand & Rozance, 2017). However islet insulin content was unaffected by chronic norepinephrine administration, which indicates independent pathways between synthesis and secretion (Chen et al., 2017). We have shown that insulin content and mRNA expression are lower in FGR $\beta$-cells, which lowers the readily releasable pool of insulin (Kelly et al., 2017; Limesand et al., 2005, 2006). We speculate that there may still be a lower rate of insulin synthesis because increases in both $\beta$-cell number and $\beta$-cell insulin content are required for optimal rates of insulin synthesis (Limesand et al., 2006). Therefore lower GPAIS is predicted to be caused by less insulin content and reduced $\beta$-cell mass despite higher $\beta$-cell proliferation rates. However this point remains unvalidated and is a limitation of the current study.

It is established that $\beta$-cell exposure to hypoxic conditions causes dysfunction resulting in impaired insulin secretion (Gerber & Rutter, 2017; Gunton, 2020; Sato et al., 2014; Tsuyama et al., 2023). The positive association of glucose-stimulated insulin concentrations and $PaO_2$ in FAS fetuses shows that hypoxemia reduces $\beta$-cell function below a threshold of $\sim$18 mmHg because no correlation was identified in FOG or CON fetuses. Basal insulin concentrations also showed a positive relationship with basal glucose but were not different between experimental groups (pooled slope 0.66 and intercept $-0.18$). We conclude that $PaO_2$ levels must remain above the critical threshold; otherwise oxygen becomes a limiting factor for insulin secretion in FGR fetuses.

In conclusion we have shown that the supplementation of oxygen and glucose has potential therapeutic benefits in fetal sheep with placental insufficiency-induced FGR. Moreover we have answered a critical question by showing that FGR even in late gestation is reversible, or at least some aspects of growth are reversible with nutrient interventions. Here we report that the combined treatment with oxygen and glucose increased growth in FGR fetuses, which was supported by improved thoracic linear growth rates and higher rates of proliferation in pancreatic $\beta$-cells and skeletal muscle satellite cells. Furthermore there were no significant differences in plasma amino acid concentrations between the FOG and FAS groups, which supports the hypothesis that glucose supplementation with adequate oxygenation will realign amino acid utilization for protein accretion rather than providing substrates for oxidative metabolism.

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

## Additional information

### Data availability statement

The data are available from the corresponding author upon reasonable request.

### Competing interests

None declared.

### Author contributions

All experiments were performed at the University of Arizona. The authors have contributed to the conception of the work (M.V., S.W.L.), the acquisition, analysis or interpretation of data for the work (M.V., D.B.C., E.H.J., R.I.L.R., A.R.T., S.K., W.Z., M.J.A., L.D.B., S.W.L.), drafting of the work (M.V., D.B.C.) and revising it critically for important intellectual content (M.V., D.B.C., E.H.J., R.I.L.R., A.R.T., S.K., W.Z., M.J.A., L.D.B., S.W.L.). All authors approved the final version of the manuscript.

### Funding

This work was supported by the National Institutes of Health RO1 DK084842, R01HD079404 (L.D.B.), S10OD023553 (L.D.B.) and the U.S. Department of Agriculture's National Institute of Food and Agriculture Award # 2023-67015-40838.

### Acknowledgements

We are grateful to the staff at the University of Arizona's Agricultural Research Center who assisted with the animal care and surgery.

## Keywords

fetal therapy, insulin secretion, intrauterine growth restriction, intrauterine intervention, pancreatic $\beta$-cell, skeletal muscle myoblast

## Supporting information

Additional supporting information can be found online in the Supporting Information section at the end of the HTML view of the article. Supporting information files available:

**Peer Review History**

