## [Peer Review History · The Journal of Physiology]

Oxygen and Glucose Supplementation Increases Rates of Linear Growth and Cell Proliferation in Growth Restricted Fetal Sheep

Mariangel Varela, Daniel B. Chrisenberry, Eliza H. Johnson, Rosa I. Luna-Ramirez, Ayna R. Tracy, Sanya Kathuria, Weicheng Zhao, Miranda J. Anderson, Laura Davidson Brown, and Sean W. Limesand
DOI: 10.1113/JP290141

Corresponding author(s): Sean Limesand (limesand@arizona.edu)

The following individual(s) involved in review of this submission have agreed to reveal their identity: Simerdeep Kaur Dhillon (Referee #2)

Review Timeline:	Submission Date:	15-Sep-2025
	Editorial Decision:	13-Oct-2025
	Revision Received:	19-Jan-2026
	Accepted:	05-Feb-2026

Senior Editor: Laura Bennet

Reviewing Editor: Janna Morrison

Transaction Report:

Re: JP-RP-2025-290141 "Oxygen and Glucose Supplementation Increases Rates of Linear Growth and Cell Proliferation in Growth Restricted Fetal Sheep" by Mariangel Varela, Daniel B. Chrisenberry, Eliza H. Johnson, Rosa I. Luna-Ramirez, Ayna R. Tracy, Sanya Kathuria, Weicheng Zhao, Miranda J. Anderson, Laura Davidson Brown, and Sean W. Limesand

Dear Dr Limesand,

Thank you for submitting your manuscript to The Journal of Physiology. It has been assessed by a Reviewing Editor and by 2 expert referees and we are pleased to tell you that it is potentially acceptable for publication following satisfactory major revision.

Please address all the points raised and incorporate all requested revisions or explain in your Response to Referees why a change has not been made. We hope you will find the comments helpful and that you will be able to return your revised manuscript within 2 months. If your article is NOT for a Special Issue, you may have 9 months to revise. If you require an extension, please contact journal staff: jp@physoc.org. Please note that this letter does not constitute a guarantee for acceptance of your revised manuscript.

REVISION CHECKLIST:

We look forward to receiving your revised submission.

Yours sincerely,

Laura Bennet
Senior Editor
The Journal of Physiology

REQUIRED ITEMS

- 1) - Include a Key Points list in the article itself, before the Abstract.
 - 2) - Author photo and profile. First or joint first authors are asked to provide a short biography (no more than 100 words for one author or 150 words in total for joint first authors) and a portrait photograph. These should be uploaded and clearly labelled together in a Word document with the revised version of the manuscript. See Information for Authors for further details.
 - 3) - The contact information for the person responsible for 'Research Governance' at your institution needs to be provided. This includes their name and an institutional email address. Please ensure the contact is not an author on this paper and provide an alternate contact if necessary, or confirm in the submission form that the author whose email was provided has sole responsibility for research governance. This is the person who is responsible for regulations, principles and standards of good practice in research carried out at the institution, for instance the ethical treatment of animals, the keeping of proper experimental records or the reporting of results.
 - 4) - You must start the Methods section with a paragraph headed Ethical approval (https://jp.msubmit.net/cgi-bin/main.plex?form_type=display_requirements#methods).
- Research must comply with The Journal's policies regarding animal experiments (<https://physoc.onlinelibrary.wiley.com/hub/animal-experiments>) and adherence to these policies must be stated in the manuscript.
- Authors should confirm in their Methods section that their experiments were carried out according to the guidelines laid down by their institution's animal welfare committee, including an ethics approval reference number. The Methods section must contain a statement about access to food, water and housing, details of the anaesthetic regime: anaesthetic used, dose and route of administration, and method of killing the experimental animals.
- 5) - Your manuscript must include a complete Additional Information section, including competing interests; funding; author contributions and acknowledgements.
 - 6) - Please ensure that any tables are editable and in Word format, and wherever possible, embedded in the article file itself.
 - 7) - Please ensure that the Article File you upload is a Word file.
 - 7) - Papers must comply with the Statistics Policy: <https://jp.msubmit.net/cgi-bin/main.plex?>

form_type=display_requirements#statistics.

In summary:

- If n {less than or equal to} 30, all data points must be plotted in the figure in a way that reveals their range and distribution. A bar graph with data points overlaid, a box and whisker plot or a violin plot (preferably with data points included) are acceptable formats.
- If $n > 30$, then the entire raw dataset must be made available either as supporting information, or hosted on a not-for-profit repository, e.g. FigShare, with access details provided in the manuscript.
- 'n' clearly defined (e.g. x cells from y slices in z animals) in the Methods. Authors should be mindful of pseudoreplication.
- All relevant 'n' values must be clearly stated in the main text, figures and tables.
- The most appropriate summary statistic (e.g. mean or median and standard deviation) must be used. Standard Error of the Mean (SEM) alone is not permitted.
- Exact p values must be stated. Authors must not use 'greater than' or 'less than'. Exact p values must be stated to three significant figures even when 'no statistical significance' is claimed.

9) - A Data Availability Statement is required for all papers reporting original data. This must be in the Additional Information section of the manuscript itself. It must have the paragraph heading 'Data Availability Statement'. All data supporting the results in the paper must be either: in the paper itself; uploaded as Supporting Information for Online Publication; or archived in an appropriate public repository. The statement needs to describe the availability or the absence of shared data. Authors must include in their statement: a link to the repository they have used, or a statement that it is available as Supporting Information; reference the data in the appropriate section(s) of their manuscript; and cite the data they have shared in the References section. Whenever possible, the scripts and other artefacts used to generate the analyses presented in the paper should also be publicly archived. If sharing data compromises ethical standards or legal requirements then authors are not expected to share it, but must note this in their statement. For more information, see our Statistics Policy.

10) - Please include an Abstract Figure file, as well as the Figure Legend text within the main article file. The Abstract Figure is a piece of artwork designed to give readers an immediate understanding of the research and should summarise the main conclusions. If possible, the image should be easily 'readable' from left to right or top to bottom. It should show the physiological relevance of the manuscript so readers can assess the importance and content of its findings. Abstract Figures should not merely recapitulate other figures in the manuscript. Please try to keep the diagram as simple as possible and without superfluous information that may distract from the main conclusion(s). Abstract Figures must be provided by authors no later than the revised manuscript stage and should be uploaded as a separate file during online submission labelled as File Type 'Abstract Figure'. Please also ensure that you include the figure legend in the main article file. All Abstract Figures should be created using BioRender. Authors should use The Journal's premium BioRender account to export high-resolution images. Details on how to use and access the premium account are included as part of this email.

EDITOR COMMENTS

Senior Editor:

When revising your manuscript please ensure you fully address all of the reviewers comments and you add information that is required by our ethics guidelines.

Reviewing Editor:

This study takes an interesting approach at replacing oxygen and glucose in a preclinical model of placental insufficiency. These are challenging studies to complete.

Methods Details:

Details of humane killing and tissue collection.

Please include ethics number in methods. Please ensure that all details of animal work are outlined in the paper as per the Journal guidelines. Please explain why penicillin and Ketofen were administered. Please provide details of humane killing of

the animals and postmortem.

Please show the control group first.

Please add the P values for the one way ANOVA to Table 2.

Figure 1. Please provide definitions in the legend for the abbreviations used in the figure.

Please include the P values for the ANOVAs on Figure 2. 2 way ANOVA = P_{treatment}, P_{time} and P_{interaction}.

For PO₂, please use subscript for 2. In figure 4C, please include the equation of the line, P and r² for each group.

Figure 5A is cut off on the right side.

Please include the information required under Additional Information as per the journal guidelines.

REFeree COMMENTS

Referee #1:

Varela and coauthors have evaluated impacts of a 10-day period of oxygen and glucose supplementation on growth, circulating concentrations of anabolic hormones and metabolites and in vivo glucose-stimulated insulin secretion in severely growth-restricted fetal sheep. This work builds upon the group's previous reports of improved in vivo glucose-stimulated insulin secretion and insulin sensitivity after a 5-day period of supplementation in the same FGR model. This is an elegant approach to normalise fetal oxygen and glucose abundance. In the present study, with the longer duration of treatment, the authors have demonstrated partial restoration of growth and anabolic hormone abundance, and restoration of glucose-stimulated but not amino acid-stimulated insulin secretion by the FGR fetus. This work is exciting as it opens new potential therapeutic options to improve fetal growth in pregnancies complicated by FGR, but I feel that rationale, data presentation and interpretation could each be improved. Please see suggestions below.

Major points for revision:

1. Novelty: In the abstract and elsewhere in the manuscript, please explain what is different about this study compared to your group's previous work using oxygen and glucose supplementation in the FGR sheep fetus. The novelty is not clear as written. For example, GSIS is also reported in the previous study. Explaining the differences/gap remaining from the Camacho et al 2022 paper is really needed to understand the value of the present study, which I think are mostly around extended treatment with sustained benefits, faster growth and partial restoration of anabolic hormone abundance.
2. Introduction (paragraph beginning line 70): The first sentence implies that no previous studies have been conducted, which seems misleading as written. Additional detail and discussion are needed of existing literature and prior approaches that have been used to promote fetal growth in the setting of FGR. For example, several studies of anabolic hormone supplementation have partially restored growth rates of FGR fetal sheep (e.g. de Boo et al. 2008 AJOG 199:559.e1-559.e9). See also the Wali et al 2012 paper which is in your reference list but not cited here in relation to variable responses to interventions in FGR fetuses. Unpacking these separately into studies supplying oxygen or nutrients and studies in which anabolic hormones are given might be useful in explaining the variable effects. For studies of single nutrient supplementation, what is meant by "fetal toxicity"? Acidemia? Hypoxia? Death?
3. Methods, fetal sheep preparation: Please include citations to previous studies in which the methods have been used previously, including for the maternal hyperthermia model of FGR. It may be possible to edit this section for length by citing these previous studies, as well as your group's previous study of oxygen and glucose supplementation in this model.
4. Statistical analyses and results presentation: The description of the statistical methods and the data presentation are confusing and it is not clear that the comparisons are appropriate. Analysis of changes over time or of pre- and treatment values should not use a simple 2-way ANOVA - these are repeat measures on the same animals and are not independent. It is not entirely clear what values are used for the "treatment" values throughout, e.g. the bars labelled "T" in Figure 2. Is this the final value (which would seem appropriate if abundance changes gradually during treatment) or an average of all measures made during treatment, or something else? This confusion is exacerbated by description of both what appears to be time-series data (changes over time within each group, using daily or more frequent data) with pre-post comparisons, within the same sentence (as noted above, it is not always clear what the "T" or "post" values actually are). I strongly recommend revising the statistical analysis, to describe changes across time during treatment in one sentence and pre vs during treatment outcomes separately. I also suggest that the "treatment" values should be the final concentrations obtained, on day 10 of treatment (except for GSIS & GPAIS, linear growth rates and cellular proliferation results, where there is a single measure per fetus). The methods for the statistical analyses would also benefit from rewriting for clarity,

including clearly stating what factors are being evaluated. I suggest using the word "condition" or similar (pre vs treatment) to avoid confusion with group (CON, FAS, FOG). "Treatment" is often used to describe groups, rather than time or before/after factors. This is confusing currently, e.g. line 331 where both group and treatment are used.

5. Data presentation, figures: Within graph panels for profiles across time, such as Fig 2 left-hand panels, please include P-values for the factors that are being analysed (group, day, interaction?). Similarly, for graph panels showing pre-treatment and during treatment values for individual animals with group means and variance, please include P-values for the factors that are being analysed (group, condition, interaction?). Data in J Physiol must be presented as mean +/- standard deviation (not standard error) for normally distributed data - this should also be stated in the statistical analysis section.

6. Post-hoc comparisons: It is not appropriate to perform post-hoc tests between all groups and in both conditions (for example for pre and during Tx for leucine, phenylalanine and alanine concentrations in Table 1). Only where an interaction between group and time (pre/post) is significant should post-hoc tests be used to assess effects of time within each group. Where group effects are significant without an interaction, then post-hoc tests should compare groups for differences (regardless of time) and these are probably most clearly reported using a numeral next to the "group" p-value with group differences reported in the corresponding footnote. This issue applies to all presentation of post-hoc tests, where it appears that the six combinations of treatment and time/condition have been compared using pair-wise tests (similar issues in Figures 2, 3, 4).

7. Results, fetal well-being indicators: Particularly in the context of a study that aims to improve fetal growth in FGR without fetal compromise, addition of pH and acid-base excess data that should be available from blood gas measures is needed to provide evidence for positive effects of oxygen plus glucose without deleterious effects on fetal well-being.

8. Figure 4 C: Which of these correlations are significant? Please add R and P values for each line to the panel. This affects conclusions based on these relationships, and the corresponding section of discussion about a threshold oxygen availability for GSIS (lines 420-).

9. Discussion: Editing of the manuscript to more clearly explain the relationships between the present study and past work would improve the discussion. The importance of the present results and/or their implications such as for development of an effective intervention for FGR in human pregnancies is not clearly explained.

10. Discussion, lines 438-442: It is not clear which sources support which part of the statement. The interpretation does not appear correct - in fact several of these studies report accelerated neonatal growth after environmental constraints causing fetal growth restriction are removed, and catch-up growth in the first months of life are characteristics of human SGA babies also. The sentence needs clarification to explain what you mean by "faster growth rates in FOG" - compared to what? Pre-treatment? FAS group?

11. Discussion, lines 446-451: Growth may occur through proliferation or hypertrophy of cells, with different functional impacts on later life. This point may be worth considering for discussion. What do you think the mechanisms driving higher proliferation rates in beta-cells and muscle satellite cells are?

12. Discussion, lines 466-469: Please clarify which statements relate to the present study and which are descriptions of previously published work throughout the discussion. This statement does not appear to reflect results of the present study, as insulin was not higher in FOG compared to FAS during treatment and it does not appear that insulin increased in the FOG group during compared to pre treatment values (Figure 2E).

13. Discussion, lines 499-: This discussion of amino acid concentrations is confusing as it is often unclear whether the statements relate to pre-post comparisons, changes across treatment, or present/previous work.

14. Discussion, lines 515-: This section would be clearer if you start with a discussion of results of the present study rather than past work.

15. Discussion, lines 543-: This section might fit better earlier in the discussion, where you describe efficacy of the intervention in normalising fetal nutrient and oxygen availability.

Minor suggestions:

16. Abstract:

a. Additional methods information would be helpful in the abstract - key points of the supplementation approach as well as numbers per group.

b. Please explain what is meant by "improved glucose homeostasis" (lines 28-30). Does this mean glucose tolerance?

c. The conclusion repeats the results and would be improved by interpreting the significance of the present study. Does it have clinical implications? How does this new study add to existing knowledge?

17. Introduction, line 78: I suggest replacing "Alternatively" with "We therefore", assuming that the oxygen-glucose supplementation approach in Camacho is based on the limited efficacy of these previous approaches?
18. Introduction, aims: In the final sentences describing aims of the present study, is the effect of treatment on circulating concentrations of anabolic hormones and nutrients a key part of your aim? If so, consider including here.
19. Introduction, line 85: Should this be "during" rather than "after", since the growth measures are continuous throughout the treatment period?
20. Methods, Ethical approval: Please see instructions to authors and include statement regarding compliance with journal requirements. Include the specific approval number.
21. Methods, Experimental design, minor suggestions: Lines 128-9: Incomplete sentence. Line 144 - What oxygen concentration was supplied to the ewe. Is this pure oxygen? Line 147 - Suggest adding "fetal" before PaO₂ to clarify which animal is being targeted. Lines 151-153 - Were air insufflation and saline infusion rates in the FAS group matched to those of the FOG group, or at fixed rates? Line 153 - "placed" not "place" Lines 160-161 - needs clarification. Am I interpreting this correctly as "morning and afternoon values were averaged to obtain a daily value, which was plotted against time and a linear regression fitted to obtain growth rate during treatment"? Line 163 - space before units.
22. Methods, Fetal GSIS Study: Line 174 - please clarify - are these historical cohorts of control and FGR fetuses and are doses different between the two groups? Did you calculate the doses per kg based on fetal weight at necropsy? Line 179 - "has been shown to elicit" can be written more succinctly as "elicits".
23. Methods, Histological analyses: The level of detail in this section is excessive. Provide citation/s to previous studies that have used these methods and a brief overview. It is not explicit from the description of slide relabelling whether cell counts were performed blinded and this needs to be clarified. The description of proliferating cell proportion calculations for satellite cells is hard to follow - this was clearer for beta-cells and I suggest adapting that description.
24. Statistics, non-analysis of sex effects: Two different reasons are given (lines 258-260 and lines 272-274) - the numbers are a sufficient reason and the second sentence likely creates confusion.
25. Results, lines 280-281: Better to describe this as oxygen and glucose supplementation as the intervention is not one where maternal oxygen is given in isolation.
26. Discussion, line 420: "as previously shown" in what course?
27. Discussion, line 457: typo - relative
28. Discussion, line 462: Do you mean alleviate inhibition of?
29. References, incomplete citations: The Spiroski paper was published in 2018 - volume, issue and page numbers are also missing. In addition, article/page numbers are missing for several sources:
- Brown et al 2024 JCI Insight
 - Romagnoli et al 2021 Int J Mol Sci
 - Stremming et al 2024 J Endocrinology
30. Figure 1: It is not obvious as drawn that the ewes are subject to hyperthermia between day 40 and day 85 of pregnancy. What units are the numbers of the x axis? dGA is defined in the legend but the abbreviation has not been used within the figure. Define all abbreviations used in the timeline, i.e. GSIS and GPAIS.
31. All data figures: The control group is conventionally shown on the left-hand side of figures.
32. Figure legends, Figures 2-4: Please state when the "treatment" condition data was obtained, so that these figures stand alone.
33. Figure 3: For ease of understanding, I suggest reporting the untransformed data for norepinephrine concentrations, and stating in the footnote that data was transformed for analysis.
34. Figure 5: Suggest replacing "for" with "of" in the figure title. What is the girth increment shown relative to? Day of surgery? Why is this positive even before you start treatments? It would better reflect the effect of treatment if you report girth increment in mm from the start of treatment.
35. Percentages in figure axes: Please specify what the units are relative to. Figure 6 - % of all beta-cells? % of islet area? Figure 7 - % of Pax7+ cells?

Referee #2:

Varela and colleagues present a well-executed study investigating oxygen and glucose supplementation as a therapeutic strategy for fetal growth restriction (FGR) resulting from placental inefficiency induced by environmental hyperthermia. Their findings show that 10 days of supplementation in near-term fetuses led to improvements in several markers of FGR, including elevated insulin and IGF-1 levels, enhanced glucose-stimulated insulin secretion, increased thoracic circumference growth, and greater proliferation of β cells and satellite cells. However, fetal weight remained unaffected.

The study is thorough and well-articulated. However, I have some major concerns:

1. The rationale for initiating treatment at near-term gestation in a model characterized by relatively severe early-onset FGR should be addressed. Clarifying this choice would strengthen the translational relevance of the findings.
2. The authors have previously reported similar outcomes following a 5-day supplementation regimen at near-term. Given that the only variable altered in the current study is the duration of treatment, it would be valuable to discuss why extending the intervention to 10 days did not yield more pronounced effects.
3. Glucose infusion directly into the fetus is a highly invasive therapeutic approach. A discussion on the feasibility and potential clinical utility of this method.
4. The potential for adverse effects associated with prolonged glucose infusion and oxygen supplementation, such as hyperglycaemia and hyperoxia should be addressed.
5. Minor comment: The symbols used to denote statistical differences between groups in the postmortem data table are unclear. These should be clearly defined in the figure legend to ensure proper interpretation.

END OF COMMENTS

Senior Editor:

When revising your manuscript please ensure you fully address all the reviewers' comments and you add information that is required by our ethics guidelines.

We appreciate the thoughtful comments from the editor and reviewers and their critical review of our manuscript. We have attempted to answer all the queries and made the necessary changes to address all comments raised in the prior review.

The reviewer comments are presented in black text, and our responses are written in blue text.

Reviewing Editor:

This study takes an interesting approach at replacing oxygen and glucose in a preclinical model of placental insufficiency. These are challenging studies to complete.

We appreciate your recognition for the complexity of these experiments in fetal sheep.

Methods Details:

Details of humane killing and tissue collection.

We have added additional necropsy information on lines 156-160.

Please include ethics number in methods. Please ensure that all details of animal work are outlined in the paper as per the Journal guidelines. Please explain why penicillin and Ketofen were administered. Please provide details of humane killing of the animals and postmortem.

We added the protocol # and provided the rationale for penicillin and ketofen (lines 144-147).

Please show the control group first.

We have revised all the figures and tables to present the control group on the left.

Please add the P values for the one way ANOVA to Table 2.

The p-values were added to table 2.

Figure 1. Please provide definitions in the legend for the abbreviations used in the figure.

The definitions are included for the abbreviations.

Please include the P values for the ANOVAs on Figure 2. 2 way ANOVA = P_{treatment}, P_{time} and P_{interaction}.

We have revised Fig 2 to include p-values for the mixed model two-way ANOVA.

For PO₂, please use subscript for 2. In figure 4C, please include the equation of the line, P and r² for each group.

The title is corrected. The equations for the lines are presented in the figure and the R² and P values are presented in the figure legend.

Figure 5A is cut off on the right side.

We have corrected the figure.

Please include the information required under Additional Information as per the journal guidelines.

We have addressed the Additional Information section to meet the guidelines.

REFEREE COMMENTS

Referee #1:

Varela and coauthors have evaluated impacts of a 10-day period of oxygen and glucose supplementation on growth, circulating concentrations of anabolic hormones and metabolites and in vivo glucose-stimulated insulin secretion in severely growth-restricted fetal sheep. This work builds upon the group's previous reports of improved in vivo glucose-stimulated insulin secretion and insulin sensitivity after a 5-day period of supplementation in the same FGR model. This is an elegant approach to normalise fetal oxygen and glucose abundance. In the present study, with the longer duration of treatment, the authors have demonstrated partial restoration of growth and anabolic hormone abundance, and restoration of glucose-stimulated but not amino acid-stimulated insulin secretion by the FGR fetus. This work is exciting as it opens new potential therapeutic options to improve fetal growth in pregnancies complicated by FGR, but I feel that rationale, data presentation and interpretation could each be improved. Please see suggestions below.

Major points for revision:

1. Novelty: In the abstract and elsewhere in the manuscript, please explain what is different about this study compared to your group's previous work using oxygen and glucose supplementation in the FGR sheep fetus. The novelty is not clear as written. For example, GSIS is also reported in the previous study. Explaining the differences/gap remaining from the Camacho et al 2022 paper is really needed to understand the value

of the present study, which I think are mostly around extended treatment with sustained benefits, faster growth and partial restoration of anabolic hormone abundance.

We have reworked the abstract and introduction to better capture the scientific value and novelty of this study. As suggested, this study is meant to emphasize the intervention effect on growth and anabolic hormone response and show amino acids are not limited. We also show that there is a threshold level for PaO₂ to promote insulin secretion during the GSIS.

2. Introduction (paragraph beginning line 70): The first sentence implies that no previous studies have been conducted, which seems misleading as written. Additional detail and discussion are needed of existing literature and prior approaches that have been used to promote fetal growth in the setting of FGR. For example, several studies of anabolic hormone supplementation have partially restored growth rates of FGR fetal sheep (e.g. de Boo et al. 2008 AJOG 199:559.e1-559.e9). See also the Wali et al 2012 paper which is in your reference list but not cited here in relation to variable responses to interventions in FGR fetuses. Unpacking these separately into studies supplying oxygen or nutrients and studies in which anabolic hormones are given might be useful in explaining the variable effects. For studies of single nutrient supplementation, what is meant by "fetal toxicity"? Acidemia? Hypoxia? Death?

We apologize for our over generalization of the FGR capacity for growth reversal. We have reorganized this paragraph to capture limitations that are specific for oxygen and nutrient interventions and highlight the need for our combination approach with oxygen and glucose. We have expanded our discussion to capture the benefits of IGF-1 administration found in the reports indicated.

3. Methods, fetal sheep preparation: Please include citations to previous studies in which the methods have been used previously, including for the maternal hyperthermia model of FGR. It may be possible to edit this section for length by citing these previous studies, as well as your group's previous study of oxygen and glucose supplementation in this model.

We have edited the methods section as recommended, while maintaining the Journal's Rigour and Reproducibility requirements for animal experiments.

4. Statistical analyses and results presentation: The description of the statistical methods and the data presentation are confusing and it is not clear that the comparisons are appropriate. Analysis of changes over time or of pre- and treatment values should not use a simple 2-way ANOVA - these are repeat measures on the same animals and are not independent. It is not entirely clear what values are used for the "treatment" values throughout, e.g. the bars labelled "T" in Figure 2. Is this the final value (which would seem appropriate if abundance changes gradually during treatment)

or an average of all measures made during treatment, or something else? This confusion is exacerbated by description of both what appears to be time-series data (changes over time within each group, using daily or more frequent data) with pre-post comparisons, within the same sentence (as noted above, it is not always clear what the "T" or "post" values actually are). I strongly recommend revising the statistical analysis, to describe changes across time during treatment in one sentence and pre vs during treatment outcomes separately. I also suggest that the "treatment" values should be the final concentrations obtained, on day 10 of treatment (except for GSIS & GPAIS, linear growth rates and cellular proliferation results, where there is a single measure per fetus). The methods for the statistical analyses would also benefit from rewriting for clarity, including clearly stating what factors are being evaluated. I suggest using the word "condition" or similar (pre vs treatment) to avoid confusion with group (CON, FAS, FOG). "Treatment" is often used to describe groups, rather than time or before/after factors. This is confusing currently, e.g. line 331 where both group and treatment are used.

We have reworked the statistical analysis section to explain the procedures for each analysis. As suggested, we have used "condition" instead of "treatment" to identify the pre-/ intervention periods in our comparison and separated their descriptions.

For figure 2: daily measurements over time were compared using mixed ANOVA procedures with fetus as the random effect. Main effects were experimental group (CON, FAS, and FOG), day of experimental intervention, and group x day interaction.

Responses between experimental conditions (intervention) were analyzed by comparing the pre-intervention values and the average values during the intervention (average days 1-10). We chose to average the entire intervention period because our previous work showed chronic not acute supplementation strategies are required for improve glucose tolerance. Furthermore, these comparisons highlight the fetal response (or lack of a response) throughout the intervention. Mixed ANOVA procedures were used with fetus as the random effect and experimental group (CON, FAS, and FOG), experimental condition (pre-intervention and intervention) and their interaction as the main effects.

In figure 2, we have chosen to present our data analysis for the interventions in two ways to illustrate stable clamps during the intervention to compare interventions means with pre-intervention starting points.

The results have been revised to reflect the terms for each analysis.

5. Data presentation, figures: Within graph panels for profiles across time, such as Fig 2 left-hand panels, please include P-values for the factors that are being analysed (group, day, interaction?). Similarly, for graph panels showing pre-treatment and during treatment values for individual animals with group means and variance, please include

P-values for the factors that are being analysed (group, condition, interaction?). Data in J Physiol must be presented as mean +/- standard deviation (not standard error) for normally distributed data - this should also be stated in the statistical analysis section.

We have added the p-values for the main effects to all figures and indicated the values are means +/- SD.

6. Post-hoc comparisons: It is not appropriate to perform post-hoc tests between all groups and in both conditions (for example for pre and during Tx for leucine, phenylalanine and alanine concentrations in Table 1). Only where an interaction between group and time (pre/post) is significant should post-hoc tests be used to assess effects of time within each group. Where group effects are significant without an interaction, then post-hoc tests should compare groups for differences (regardless of time) and these are probably most clearly reported using a numeral next to the "group" p-value with group differences reported in the corresponding footnote. This issue applies to all presentation of post-hoc tests, where it appears that the six combinations of treatment and time/condition have been compared using pair-wise tests (similar issues in Figures 2, 3, 4).

We have revised the presentation for Table 1 by splitting the table into two tables. The new table 1 only presents the amino acids with a group x condition interaction and the new table 2 presents the group means (+/- SD). Standard deviations are presented in all other figures except Figure 4D where 95% CI are used to represent the data as indicated in the Journal's guidelines.

7. Results, fetal well-being indicators: Particularly in the context of a study that aims to improve fetal growth in FGR without fetal compromise, addition of pH and acid-base excess data that should be available from blood gas measures is needed to provide evidence for positive effects of oxygen plus glucose without deleterious effects on fetal well-being.

We have added the pH, acid-base excess data to Figure 2.

8. Figure 4 C: Which of these correlations are significant? Please add R and P values for each line to the panel. This affects conclusions based on these relationships, and the corresponding section of discussion about a threshold oxygen availability for GSIS (lines 420-).

We have included the R and P values in the legend of figure 4C.

9. Discussion: Editing of the manuscript to more clearly explain the relationships between the present study and past work would improve the discussion. The importance of the present results and/or their implications such as for development of an effective intervention for FGR in human pregnancies is not clearly explained.

We have added an introductory paragraph to summarize the major findings from this and the previous report. Additionally we highlight the potential clinical benefits for OG supplementation.

10. Discussion, lines 438-442: It is not clear which sources support which part of the statement. The interpretation does not appear correct - in fact several of these studies report accelerated neonatal growth after environmental constraints causing fetal growth restriction are removed, and catch-up growth in the first months of life are characteristics of human SGA babies also. The sentence needs clarification to explain what you mean by "faster growth rates in FOG" - compared to what? Pre-treatment? FAS group?

We have revised this statement to reflect differences between fetal responses and omitted the postnatal statements (lines 450-469).

11. Discussion, lines 446-451: Growth may occur through proliferation or hypertrophy of cells, with different functional impacts on later life. This point may be worth considering for discussion. What do you think the mechanisms driving higher proliferation rates in beta-cells and muscle satellite cells are?

We targeted proliferation instead of hypertrophy in the pancreas because we know cell numbers are reduced but β -cell area is unaffected. As for the mechanism in the pancreas we propose that oxygen and glucose have a combined action to stimulate proliferation.

We target proliferation in the satellite cells because in the fetus cell hypertrophy is associated with myonuclei domain, which requires satellite cell proliferation and differentiation. However, the latter was not measured directly. We propose that the mechanism of action for increased satellite cell proliferation is IGF-1, because IGF-1 has been shown previously to stimulate proliferation.

12. Discussion, lines 466-469: Please clarify which statements relate to the present study and which are descriptions of previously published work throughout the discussion. This statement does not appear to reflect results of the present study, as insulin was not higher in FOG compared to FAS during treatment and it does not appear that insulin increased in the FOG group during compared to pre treatment values (Figure 2E).

We have revised the statement to reflect the current data for IGF-1.

13. Discussion, lines 499-: This discussion of amino acid concentrations is confusing as it is often unclear whether the statements relate to pre-post comparisons, changes across treatment, or present/previous work.

We have reworked the discussion on amino acids to clarify our major points.

14. Discussion, lines 515-: This section would be clearer if you start with a discussion of results of the present study rather than past work.

We have expanded our explanation for the current findings (lines 540-544).

15. Discussion, lines 543-: This section might fit better earlier in the discussion, where you describe efficacy of the intervention in normalising fetal nutrient and oxygen availability.

We moved this paragraph up to line 503.

Minor suggestions:

16. Abstract:

a. Additional methods information would be helpful in the abstract - key points of the supplementation approach as well as numbers per group.

We have followed the Journal's abstract guidelines on essential and nonessential elements, while addressing your recommendation to revise the abstract.

b. Please explain what is meant by "improved glucose homeostasis" (lines 28-30). Does this mean glucose tolerance?

Yes. We have revised this statement.

c. The conclusion repeats the results and would be improved by interpreting the significance of the present study. Does it have clinical implications? How does this new study add to existing knowledge?

We have added a statement for clinical relevance.

17. Introduction, line 78: I suggest replacing "Alternatively" with "We therefore", assuming that the oxygen-glucose supplementation approach in Camacho is based on the limited efficacy of these previous approaches?

The statement is revised as recommended.

18. Introduction, aims: In the final sentences describing aims of the present study, is the effect of treatment on circulating concentrations of anabolic hormones and nutrients a key part of your aim? If so, consider including here.

We expanded our description of the study aims to reflect the new information shown.

19. Introduction, line 85: Should this be "during" rather than "after", since the growth measures are continuous throughout the treatment period?

Yes. Revised

20. Methods, Ethical approval: Please see instructions to authors and include statement regarding compliance with journal requirements. Include the specific approval number.

Yes. Revised

21. Methods, Experimental design, minor suggestions: Lines 128-9: Incomplete sentence. Line 144 - What oxygen concentration was supplied to the ewe. Is this pure oxygen? Line 147 - Suggest adding "fetal" before PaO₂ to clarify which animal is being targeted. Lines 151-153 - Were air insufflation and saline infusion rates in the FAS group matched to those of the FOG group, or at fixed rates? Line 153 - "placed" not "place" Lines 160-161 - needs clarification. Am I interpreting this correctly as "morning and afternoon values were averaged to obtain a daily value, which was plotted against time and a linear regression fitted to obtain growth rate during treatment"? Line 163 - space before units.

We added 100% O₂ and fetal; explained the rates are fixed in FAS fetuses; corrected placed; clarified the measurement statement; and added the space.

22. Methods, Fetal GSIS Study: Line 174 - please clarify - are these historical cohorts of control and FGR fetuses and are doses different between the two groups? Did you calculate the doses per kg based on fetal weight at necropsy? Line 179 - "has been shown to elicit" can be written more succinctly as "elicits".

In response to point 3 we have removed these references to prior work and provided references for the GSIS GPAIS procedures and target values.

23. Methods, Histological analyses: The level of detail in this section is excessive. Provide citation/s to previous studies that have used these methods and a brief overview. It is not explicit from the description of slide relabelling whether cell counts were performed blinded and this needs to be clarified. The description of proliferating cell proportion calculations for satellite cells is hard to follow - this was clearer for beta-cells and I suggest adapting that description.

We have revised as recommended both here and your 3rd point.

24. Statistics, non-analysis of sex effects: Two different reasons are given (lines 258-260 and lines 272-274) - the numbers are a sufficient reason and the second sentence likely creates confusion.

Corrected as recommended.

25. Results, lines 280-281: Better to describe this as oxygen and glucose supplementation as the intervention is not one where maternal oxygen is given in isolation.

Corrected

26. Discussion, line 420: "as previously shown" in what course?

The statement is revised.

27. Discussion, line 457: typo - relative

Corrected

28. Discussion, line 462: Do you mean alleviate inhibition of?

Corrected

29. References, incomplete citations: The Spiroski paper was published in 2018 - volume, issue and page numbers are also missing. In addition, article/page numbers are missing for several sources:

- a. Brown et al 2024 JCI Insight
- b. Romagnoli et al 2021 Int J Mol Sci
- c. Stremming et al 2024 J Endocrinology

All are corrected.

30. Figure 1: It is not obvious as drawn that the ewes are subject to hyperthermia between day 40 and day 85 of pregnancy. What units are the numbers of the x axis? dGA is defined in the legend but the abbreviation has not been used within the figure. Define all abbreviations used in the timeline, i.e. GSIS and GPAIS.

Revised as recommended.

31. All data figures: The control group is conventionally shown on the left-hand side of figures.

Revised as recommended.

32. Figure legends, Figures 2-4: Please state when the "treatment" condition data was obtained, so that these figures stand alone.

Revised as recommended.

33. Figure 3: For ease of understanding, I suggest reporting the untransformed data for norepinephrine concentrations, and stating in the footnote that data was transformed for analysis.

We have removed the NE graph because condition differences were not significant with the analysis.

34. Figure 5: Suggest replacing "for" with "of" in the figure title. What is the girth increment shown relative to? Day of surgery? Why is this positive even before you start

treatments? It would better reflect the effect of treatment if you report girth increment in mm from the start of treatment.

Revised as recommended.

35. Percentages in figure axes: Please specify what the units are relative to. Figure 6 - % of all beta-cells? % of islet area? Figure 7 - % of Pax7+ cells?

Revised as recommended.

Referee #2:

Varela and colleagues present a well-executed study investigating oxygen and glucose supplementation as a therapeutic strategy for fetal growth restriction (FGR) resulting from placental inefficiency induced by environmental hyperthermia. Their findings show that 10 days of supplementation in near-term fetuses led to improvements in several markers of FGR, including elevated insulin and IGF-1 levels, enhanced glucose-stimulated insulin secretion, increased thoracic circumference growth, and greater proliferation of β cells and satellite cells. However, fetal weight remained unaffected.

The study is thorough and well-articulated. However, I have some major concerns:

1. The rationale for initiating treatment at near-term gestation in a model characterized by relatively severe early-onset FGR should be addressed. Clarifying this choice would strengthen the translational relevance of the findings.

We have revised the interdictio to emphasize the rationale for beginning treatment after diagnosis to reflect greater clinical relevance.

2. The authors have previously reported similar outcomes following a 5-day supplementation regimen at near-term. Given that the only variable altered in the current study is the duration of treatment, it would be valuable to discuss why extending the intervention to 10 days did not yield more pronounced effects.

We have expanded our discussion to compare the 5-day and 10-day treatment periods and explain why differences in absolute weight were not pronounced (lines 450-460).

3. Glucose infusion directly into the fetus is a highly invasive therapeutic approach. A discussion on the feasibility and potential clinical utility of this method.

Translation of this intervention into clinic practice will be developed through maternal regulation without direct access to the fetus. However, for the sake of this proof of

principle experiments that show fetal improvements we chose to clamp fetal glucose directly because it reduces variability.

4. The potential for adverse effects associated with prolonged glucose infusion and oxygen supplementation, such as hyperglycaemia and hyperoxia should be addressed.

We have purposefully managed oxygen and glucose to achieve near normal levels in the fetus and avoid hyperoxia and hyperglycemia in the fetus. Previously (PMID: 35560217), we discussed that risks of fetal hyperoxia is low based on the fetal oxyhemoglobin dissociation curves. While we have shown the negative impact of hyperglycemia on fetal insulin secretion, these comparisons seem outside the scope of our current work. We have clarified the study parameters to avoid secondary complications.

5. Minor comment: The symbols used to denote statistical differences between groups in the postmortem data table are unclear. These should be clearly defined in the figure legend to ensure proper interpretation.

In table 3 we have included the statement “Group differences ($P < 0.05$) were determined with a Tukey-Kramer test, and the different letters indicate differences between groups” and added p-values

Dear Professor Limesand,

Re: JP-RP-2026-290141R1 "Oxygen and Glucose Supplementation Increases Rates of Linear Growth and Cell Proliferation in Growth Restricted Fetal Sheep" by Mariangel Varela, Daniel B. Chrisenberry, Eliza H. Johnson, Rosa I. Luna-Ramirez, Ayna R. Tracy, Sanya Kathuria, Weicheng Zhao, Miranda J. Anderson, Laura Davidson Brown, and Sean W. Limesand

We are pleased to tell you that your paper has been accepted for publication in The Journal of Physiology.

Yours sincerely,

Laura Bennet
Senior Editor
The Journal of Physiology

IMPORTANT POINTS TO NOTE FOLLOWING ACCEPTANCE OF YOUR PAPER:

- **IMPORTANT NOTICE ABOUT OPEN ACCESS:** To assist authors whose funding agencies mandate immediate public access to published research findings, The Journal of Physiology allows authors to pay an Open Access (OA) fee to have their papers made freely available immediately on publication.

- You can help your research get the attention it deserves! Check out Wiley's free Promotion Guide for best-practice recommendations for promoting your work at: www.wileyauthors.com/eoo/guide. You can learn more about Wiley Editing Services which offers professional video, design, and writing services to create shareable video abstracts, infographics, conference posters, lay summaries, and research news stories for your research at: www.wileyauthors.com/eoo/promotion.

- If you would like to receive our 'Research Roundup', a monthly newsletter highlighting the cutting-edge research published in The Physiological Society's family of journals (The Journal of Physiology, Experimental Physiology, Physiological Reports, The Journal of Nutritional Physiology and The Journal of Precision Medicine: Health and Disease), please click this link, fill in your name and email address and select 'Research Roundup': <https://www.physoc.org/journals-and-media/membernews>

EDITOR COMMENTS

Reviewing Editor:

Comments to the Author (Required):

Thank you for revising the paper in line with the reviewers comments. This is an interesting and complex study.

REFEREE COMMENTS

Referee #1:

Thank you for carefully addressing each point raised by the editor and referees. I have no further comments or requested changes.

Well done on completion of this challenging and important study.

Referee #2:

The authors have addressed all the previously raised concerns. I have no further comments to add.